# Validity and reliability of anthropometric equations versus Dual X-ray Absorptiometry to estimate body composition in athletes with unilateral lower-limb amputation: A pilot study

Laura Victoria Rivera-Amézquita [1,2]*, Ximena Saavedra-Bernal[1], Sofia Diaz-Moreno[1], Alejandra Tordecilla-Sanders[1], Diana Carolina Escorcia-Gomez[1,3], Diana Marcela Ramos-Caballero[1], Zdenek Svoboda[2]

1 Universidad del Rosario, School of Medicine and Health Sciences, Rehabilitation Science Research Group, Center for the Study of Physical Activity Measurement (CEMA), Bogotá, Colombia, 2 Palacky University Olomouc, Faculty of Physical Culture, Olomouc, Czech Republic, 3 Unidad de Ciencias Aplicadas, Instituto Distrital de Recreación y Deporte IDRD, Bogotá, Colombia.

* laurav.rivera@urosario.edu.co

## Abstract

### Background

Accurate assessment of body composition is essential for monitoring health and performance in high-performance athletes. While anthropometric equations are widely used in conventional sports, their validity in athletes with unilateral lower-limb amputation remains unclear due to assumptions of symmetrical fat and muscle distribution. This study aimed to assess the concurrent validity and reliability of anthropometric equations for estimating body composition in this population, using Dual-energy X-ray Absorptiometry (DEXA) as the reference method.

### Methods

A cross-sectional study was conducted with 27 athletes (22 men, 5 women; mean age 32 ± 7.4) from para-sports including athletics, swimming, triathlon, and others. Anthropometric measurements and DEXA were used to estimate body density (BD), fat percentage (FP), lean percentage (LP), lean mass (LM), and fat mass (FM). Forty-seven anthropometric equations were evaluated. Validity was assessed using Pearson or Spearman correlation coefficients, reliability with Intraclass Correlation Coefficients (ICC), and agreement through Bland-Altman analysis and Reduced Major Axis (RMA) regression.

### Results

For FP, the Hastuti equation and the ISAK 5 components model showed the closest agreement with DEXA (mean differences 0.7 ± 4.55%, and 0.32 ± 4.8%, respectively;

**Data availability statement:** The data can be accessed through the following link: https://doi.org/10.34848/RHHTEY

**Funding:** This research was funded by Universidad del Rosario (Specific grant number: IV-FCS024 Capital Semilla) The funders had no role in study design, data collection and analysis, decisions to publish, or preparation of the manuscript.

**Competing interests:** The authors have declared that no competing interests exist

ICCs > 0.83). Durnin and Womersley with Siri also showed high reliability but a greater bias (2.6 ± 3.69%). For FM, the ISAK 5 components model and Lee DH equation demonstrated acceptable agreement (mean differences −0.71 ± 3.64% and 1.04 ± 4.35%, respectively; ICCs > 0.85). For LM, the Olshvang, Chien, and Lee DH equations showed the strongest agreement with DEXA (ICCs > 0.87).

## Conclusions

The Hastuti and the ISAK 5 components model are recommended to estimate FP. For FM, we recommend the ISAK 5 components model, followed by Lee DH equations, and for LM, the Olshvang, Chien and Lee DH. Finally, we do not recommend the use of any of the Lee, Poortsman, or ISAK 5 components model equations to estimate LP in this population.

## Introduction

The assessment of body composition in high-performance athletes is a crucial process, as it provides valuable information regarding health status, athletic performance, and progression toward sport-specific goals [1,2]. Key indicators such as fat mass (FM) and lean mass (LM) are closely linked to performance, with lower FM and higher LM being associated with enhanced cardiorespiratory fitness [3] and muscular strength [4,5]. Moreover, regular monitoring of body composition provides insights into training-induced adaptations and facilitates the early detection of potential health concerns, including eating disorders, extremely changes in body mass or dehydration [6]. In the context of Paralympic sports, body composition assessment also plays a key role in evaluating athletes' health, performance, and functional capabilities [7]. This is particularly relevant for athletes with amputations, for whom body composition measurements are critical due to a heightened risk of weight gain. Following an amputation, individuals often experience a reduction in muscle mass due to a decline in physical activity levels, which also can lead to an increase in body fat percentage (FP). Research indicates that individuals who undergo amputation often experience an increase in body mass index (BMI) within the first year post-amputation [8], with obesity prevalence reported to range from 37.9% to 48% in this population [9]. Considering these risks, accurate body composition assessment in individuals with limb loss is essential for developing targeted strategies to improve overall health and reduce the risk of chronic diseases associated with increased adiposity [8].

Given the importance of assessing body composition in Paralympic athletes, there is a need for accurate and accessible assessment methods. Dual X-ray Absorptiometry (DEXA) is widely considered the gold standard for measuring body composition [10], offering high precision in estimating body fat, muscle mass, and bone mineral density [11]. Despite its high accuracy, DEXA presents several limitations that hinders its routine use in athletic environments. These include the high cost of the equipment, the need for substantial physical space and specific technical requirements, the necessity for skilled and experienced professionals to interpret the results accurately,

and its lack of portability. Collectively, these factors reduce the feasibility of employing DEXA for frequent or field-based assessment [12]. An established and validated method for estimating body composition involves the use of predictive equations derived from anthropometric indices. These anthropometry-based predictive equation utilize measurements such as skinfold thickness, circumferences, and bone diameters to estimate components of body composition, including FM and LM [13–15]. Different authors have highlighted the relevance of such equations for estimating FM in specific populations, such as athletes involved in different sports [16–18]. In addition, the quantification method -particularly skinfold measurement- is less affected by daily activities, such as food intake and changes in hydration status, as is the case for bioelectrical impedance analysis. Skinfold assessment entails the skilled use of a caliper to measure skinfold thickness at standardized anatomic sites, enabling the determination of the overall subcutaneous tissue [19]. Circumference measurements are obtained using a measuring tape at specific reference points, while bone diameters are assessed with a caliper, offering insights into skeletal dimensions. Although this method has some limitations, including a lack of population specificity, measurement errors, and an inability to differentiate between subcutaneous and visceral fat, it is cost-effective, requires minimal equipment, is portable, and can be applied in various sports settings [19]. The ease of applying this method has led to its widespread use in sports. Consequently, this method has also been used in athletes undergoing amputations. However, in athletes with unilateral amputations, anthropometric equations may be less accurate, as they assume a symmetrical distribution of fat and muscle mass [20], which may not necessarily be the case in athletes with unilateral amputation. Therefore, this study aims to compare the concurrent validity and reliability of anthropometric equations for estimating body composition in athletes with unilateral lower-limb amputation, using DEXA as the gold standard method, to evaluate their applicability in this population. The findings are intended to support the development of more accurate and population-specific guidelines for health professionals involved in body composition assessment in this group.

## Methods

This cross-sectional pilot validation study aimed to explore the concurrent validity and reliability of anthropometric equations for estimating body composition in athletes with unilateral lower-limb amputation, using DEXA as the reference method. The variables of interest were body density (BD), fat percentage (FP), lean percentage (LP), lean mass (LM) and fat mass (FM). Twenty-seven volunteer athletes with unilateral lower-limb amputation of legal age (over 18 years) affiliated with the District Institute of Recreation and Sport (IDRD) in one of its recognized leagues were included in the study. Athletes who agreed to participate were provided with a detailed explanation of the procedures and subsequently signed informed consent forms. Consent was obtained both verbally and in written form. Participants with active sports injuries or edema were excluded from the study. This study was approved by the Ethics Committee of the School of Medicine and Health Sciences, University of Rosario (Bogotá, Colombia) (approval number: DVO005 1087-CV1181).

The sample size was calculated based on an expected mean difference of 1 unit (percentage or Kilograms) and a standard deviation of the difference of 1.8, with a significance level of 0.05 and a statistical power of 0.8. The calculation indicated that a minimum of 26 participants was required [21]. Ultimately, 27 athletes were included in this pilot study, out of a total of 32 eligible individuals who met the inclusion criteria. The participants were identified and recruited through sports medicine, physiotherapy, and nutrition services provided by the IDRD. They were sent to the Applied Sciences Unit (UCAD) for the measurement session. Sociodemographic data were collected systematically after obtaining informed consent. Subsequently, anthropometric measurements were conducted, followed by a comprehensive assessment of the body composition using DEXA. Anthropometric measurements were performed first, with a 40-minute interval before DEXA scan to prevent potential interference between methods. During this interval, participants were instructed to refrain from eating or drinking to ensure consistent measurements conditions across both assessment techniques. The data collection period started on the 25th of January 2022 and finished on 2nd of March of 2022.

## Dual-Energy X-ray Absorptiometry (DEXA)

Body composition was assessed using the Dual-Energy X-ray Absorptiometry (DEXA) system (Hologic Horizon® DXA System Software APEX, version 5.6.0.7, Florida, MI, USA). The ionizing radiation dose for full-body composition evaluation was less than 0.007 mGy over 174 seconds [22]. Each examination was conducted by a trained operator proficient in equipment handling and measurement procedures. Before the measurement, both daily and weekly quality controls procedures required by the system were conducted using an anthropomorphic spine phantom. Calibration results were considered acceptable when falling within ± 1.5% of the phantom´s reference mean value. Over the calibration period, the phantom yielded a mean bone mineral density (BMD) of 1.009 g/cm² with a standard deviation of 0.001 g/cm², demonstrating high precision and stability of the device. Athletes were instructed to remove metallic accessories, dress lightly, and wear disposable gowns. Subsequently, the participants were positioned in a supine posture on the scanning table, with arms fully extended and hands in a pronated position. Emphasis was placed on ensuring that the thumbs did not contact the gluteal region, the legs were positioned in abduction, and the hips were internally rotated to allow proper visualization of the greater trochanter. These measurements yielded estimates of FP, FM, LP, and LM [23].

## Body composition measurements through anthropometry

The weight and height of each athlete without the prosthesis were measured using a bioimpedance analyzer (Tanita SC 331S) and a SECA stadiometer (213). Body Mass Index (BMI) was then calculated. During measurements period, the temperature in Bogota ranged between 12°C and 19°C, with no recorded temperatures falling below or exceeding this range. The average relative humidity was 79%. Anthropometric measurements were conducted by two level 1 anthropometrists following the restrictive profile protocols of the International Society for the Advancement of Kinanthropometry (ISAK) [24]. The recommended equipment, including a Cescorf segmometer, Lufkin anthropometric tape, anthropometric box, SECA stadiometer, professional Harpenden skinfold caliper, and Cescorf caliper, were used. Additionally, the anatomical references were marked with a pencil for anthropometric marking, according to the ISAK protocol. Alcohol, and cotton were used to clean the skin of each athlete. Two weeks before the measurement, those in charge conducted practice sessions. The relative intra-observer Technical Error of Measurement (TEM) was 0.7%, indicating high measurement precision. Inter-rater reliability, assessed using the intraclass correlation coefficient (ICC) for non-randomized fixed raters and calculated via Fisher´s z-transformation, was excellent (ICC: 0.977, 95% IC [0.805; 0.998] for the anthropometric measurements. During data collection, one evaluator agreed to perform the measurements (XSB), whereas the second evaluator recorded the results (SDM) in the order established by the ISAK protocol. In this study, measurements were taken from the side of the body without amputation. To minimize measurement errors, both evaluators assessed each anatomical point for the skinfold evaluation at least twice. All measurement errors were below the cutoff point, which is considered acceptable by the ISAK protocol. It is important to consider that the accuracy of anthropometric assessment relies on the use of predictive equations developed for specific populations, which assume a consistent distribution of body fat. However, this distribution can be influenced by various factors including age, sex, ethnicity, sport type, and the presence of a limb amputation. In athletes with unilateral amputation, standard practice involved taking measurements from the intact limb, in accordance with the procedures proposed by Lohman [25]. Currently, at the IDRD, body composition in athletes with amputations is assessed using these standard anthropometric protocols and predictive equations recommended by ISAK. This study aims to evaluate the validity of this method by comparing it with DEXA, the gold standard for body composition assessment. Following the anthropometric measurements, a total of 8 equations were applied to estimate BD [20,26–33], 17 for FP [34–51], 6 for FM [49,50,52–56], 3 for LP [48,49,57,58] and 16 for LM [48–50,53,56,57,59–64]. Table 1 summarizes the equations used for estimating the BD, FP, FM, LP, and LM. These equations were selected based on an extensive review of the scientific literature, with a focus on their relevance and applicability in field-based assessment settings; some of these equations are frequently used in athletes with amputation, despite having originally been

**Table 1. Relevant anthropometric equations to estimate BD, FP, FM, LP and LM reported in the evidence.**

| Variable | Definition | Author | Equation |
|---|---|---|---|
| Body Density | Amount of mass (weight) an individual has per unit volume of their body [46]. | Durnin and Womersley [26] | BD (masc) = 1.1765 – [0.0744 * log10 (BI+TR+SS+SI)] <br> BD (fem) = 1.1567–0.0717 * log10 (BI+TR+SS+SI)] |
| | | Forsyth and Sinning [20,27] | BD=1.1103 – (0.00168 * SS) – (0.00127 ·AB) |
| | | Katch and MacArdle [28] | BD=1.09665 – (0.00103 * TR) – (0.00056 * SS) – (0.00054 * AB) |
| | | Nagamine and Suzuki [20,29] | BD=1.0913 – (0.00116 * (TR+SS)) |
| | | Sloan [20,30] | BD=1.1043 – (0.001327 * TH) – (0.00131 * SS) |
| | | Wilmore and Behnke [20,31] | BD=1.08543 – (0.000886 * AB) – (0.0004 * TH) |
| | | Whithers [32,66] | BD=1.0988 – (0.0004 * (TR+SS+BI+SI+AB+TH+C)) |
| | | White [20,33] | BD=1.0958 – (0.00088 * SI) – (0.0006 * TH) |
| Fat percentage (FP) | Proportion of fat in the body relative to the total body weight [67]. | Eston [34,68] | FP = [0.12 * (BI+TR+SS+SI)] + [0.36 · (TH+C)] + 1.61 |
| | | Evans [35,68] | FP=8.997 + (0.24658 * (AB+TH+TR)) – (6.343 * Sex) – (1.998 * Race) <br> Sex: male=1, female=0 <br> Race: Afro descendants=1, white=0 |
| | | O'Connor [20,36] | FP = [0.272 * (TR+SI+TH)] – [0.0005 * (TR+SI+TH)] + 4.972 |
| | | Carter [37] | ∑6SF: TR, BI, SI, SS, TH, C. <br> FP (mas) = 2.585 + (0.1051* ∑6SF) <br> FP (fem) =3.5803 + (0.1548 * ∑6SF) |
| | | Faulkner [38] | ∑4SK: TR, SS, SI, AB. <br> FP (mas) = (∑4SF * 0.153) + 5.783 <br> FP (fem) = (∑4SF * 0.213) + 7.9 |
| | | Slaughter [39] | FP (fem) =1.33 * (TR+SS) – 0.013 * (TR+SS)$^2$ - 2.5 <br> FP (masc) = 1.21 * (TR+SS) – 0.008 * (TR+SSP)$^2$ –3.4 |
| | | Yuhasz [40] | ∑6SK: TR, SS, SI, AB, TH, C. <br> FP (masc) = 0.1051 * (∑6SK (mm) + 2.58 <br> FP (fem) = 0.1548 * (∑6SK (mm) + 3.58 |
| | | Minematsu [41] | FP=10.558×sex+0.069×age+0.667×BMI+0.314×WC −35.881 <br> Sex: 1 for men; 2 for women |
| | | Hastuti [42] | FP=17.026 + (0.509 * TR) + (0.342 * SI) – 5.594 * sex <br> Sex: 1 for men; 0 for women |
| | | Deurenberg [43] | FP=1.2 * BMI+0.23 * age− 10.8 * sex − 5,4 <br> Sex: 1 for men; 0 for women |

*(Continued)*

**Table 1.** (Continued)

| Variable | Definition | Author | Equation |
|---|---|---|---|
| | | Lean [44] | Equation 1, waist circumference<br>FP men = (0.567 * WC) + (0.101 * age) − 31.8<br>FP women = (0.439 * WC) + (0.221 * age) − 9.4 |
| | | | Equation 3, BMI<br>FP men = (1.33 * BMI) + (0.236 * age) − 20.2<br>FP women = (1.21 * BMI) + (0.262 * age) − 6.7 |
| | | | Equation 5, BMI and TR<br>FP men = (0.742 * BMI) + (0.950 * TR) + (0.335 * age) − 20<br>FP women = (0.730 * BMI) + (0.548 * TR) + (0.270 * age) − 5.9 |
| | | | Equation 6 log10 $\sum$4SK (BI, TR, SS, SI)<br>FP men = (30.69 * log10 $\sum$4SK) + (0.271 * age) − 39.9<br>FP women = (30.8 * log10 $\sum$4SK) + (0.274 * age) − 31.7 |
| | | Gómez-Ambrosi [45] | FP = −44.988 + (0.503 * age)<br>+ (10,689 * sex) + (3.172 * BMI) − (0.026 * $BMI^2$)<br>+ (0.181 * BMI * sex) − (0.02 * BMI * age)<br>− (0.005 * $BMI^2$ * sex) + (0.00021 * $BMI^2$ * age)<br>Sex: 0 for men; 1 for women |
| | | Siri [37,46] | FP = 100*(4.95/BD) − 4.5 |
| | | Brozek [37,47] | FP = 100* (4.57/BD) − 4.142 |
| | | ISAK 5 components model [48,49] | $\sum$6SK = TR + SS + SI + AB + TH + C<br>FP = (($\sum$6SK * 5.85) + 25.6)/ (170.18/H)³)/ BW |
| | | Lee DH [50] | FP men = 2.80 + (0.03 * age) − (0.04 * H) − (0.08 * BW) + (0.35 * WC) − (0.18 * Ac) + (0.00 * Cc) + (0.04 * THc) + (0.33 * TR) + (0.08 * SS) + race<br>0.53 for Mexican<br>0.19 for Hispanic<br>−0.79 for Afro descendants<br>0.7 for others<br>FP women = 32.75 + (0.08 * age) − (0.16 * H) + (0.10 * BW) + (0.12 * WC) − (0.05 * Ac) − (0.17 * Cc) + (0.22 * THc) + (0.27 * TR) + (0.05 * SS) + race<br>0.98 for Mexican<br>0.23 for Hispanic<br>−1.77 for Afro descendants<br>0.08 for others |
| | | Giro [51] | FP = −0.620 + (0.159 * $\sum$4SK) + (0.120 * WC)<br>$\sum$4SK = TR + SI + AB + TH |

*(Continued)*

| Variable | Definition | Author | Equation |
|---|---|---|---|
| Fat mass (FM) | It is the total amount of fat in the body, including both essential fat and storage fat [13]. | Al-Guindan [52] | FM men = 0.198 * BW + 0.478 * WC − 0.147 * H − 12.8<br>FM women = 0.789 * BW + 0.0786 * age − 0.342 * H + 24.5 |
| | | De Rose and Guimaraes [53] | FM = 0.01 * BW * ((SS + TR + SSP + AB) * 0.153 + 5.783) |
| | | ISAK 5 components model [48,49] | $\sum 6SK$ = TR + SS + SI + AB + TH + C<br>FM = (($\sum 6SK$ * 5.85) + 25.6) / $(170.18/H)^3$ |
| | | Heitmann [55] | FM men = (0.988 * BMI) + (0.242 * BW) + (0.094 * age) − 30.180<br>FM women = (0.988 * BMI) + (0.344 * BW) + (0.094 * age) − 30.180 |
| | | Lee DH [50] | FM men = −0.009 + (0.004 * age) − (0.108 * H) + (0.334 * BW) + (0.247 * WC) − (0.306 * Ac) − (0.075 * Cc) − (0.028 * THc) + (0.307 * TR) + (0.030 * SS) + race<br>0.154 for Mexican<br>−0.050 for Hispanic<br>−0.529 for Afro descendants<br>0.687 for others<br>FM women = 8.633 + (0.048 * age) − (0.166 * H) + (0.569 * BW) + (0.044 * WC) − (0.082 * Ac) − (0.183 * Cc) + (0.115 * THc) + (0.150 * TR) + (0.008 * SS) + race<br>0.357 for Mexican<br>−0.071 for Hispanic<br>−1.345 for Afro descendants<br>0.177 for others |
| | | Salamat [56] | FM = −11.938 + (1.606 * BMI) − (8.511 * sex)<br>0 for female<br>1 for men |
| Lean percentage (LP) | Amount of muscle in relation to the body weight [57] | Lee RC [57] | LP = LM/BW * 100.<br>LM = H * (0.00744 * $cAC^2$ + 0.00088 * $cTHC^2$ + 0.00441 * $cCC^2$) + 2.4 * sex − 0.048 * age + race + 7.8<br>cAC = relaxed Ac − (TR * 3.141/10)<br>cTHC = THc − (TH * 3.141/10)<br>cCC = Cc − (C * 3.141/10)<br>cCHC = CHc − (SS * 3.141/10)<br>Sex: 1 for male, 0 for female<br>Race: −2 for Asians, 1.1 for African American, 0 for white of Hispanic. |
| | | Poortmans [58] | LP = H * (0.0264 * $Ac^2$) + (0.0032 * $THc^2$) + (0.0015 * $Cc^2$) + (2.56 * sex) + (0.136 * age)<br>Sex: male = 1, female = 0 |
| | | ISAK 5 components model [48,49] | LP = [((ZMUS * 5.4) + 24.5)/$(170.18/H)^3$]/ BW<br>$\sum 5$Circumferences = (cAC + FAc + cTHC + cCC + cCHC)<br>ZMUS = (($\sum 5$Circumferences*(170.18/ H) − 207.21))/ 13.74<br>Corrections:<br>cAC = relaxed Ac − (TR * 3.141/10)<br>cTHC = THc − (TH * 3.141/10)<br>cCC = Cc − (C * 3.141/10)<br>cCHC = CHc − (SS * 3.141/10) |

(Continued)

| Variable | Definition | Author | Equation |
|---|---|---|---|
| Lean Mass (LM) | Soft tissues and muscle mass, but excludes fat and bone mass [69] | De Rose and Guimaraes [53] | $LM=BW - (FM+BM+RM)$<br>Fat mass $(FM) = 0.01 * BW * ((SS+TR+SSP+AB) * 0.153 + 5.783)$<br>Bone mass $(BM) = 3.02 * (H^2/100 * WD/100 * FD/100 * 400)^{0.712}$<br>Residual mass $(RM) = BW * 0.241$ |
| | | Heymsfield [53,59] | $LM=H * (0.284+0.0029 * cAC)$<br>$cAC$ men $= ((Ac - (\pi * TR))2/4 * \pi) - 10$<br>$cAC$ men women $= ((Ac - (\pi * TR))2/4 * \pi) - 6.5$ |
| | | Lee RC 1 [57] | $LM=H * (0.00744 * cAC^2 + 0.00088 * cTHC^2 + 0.00441 * cCC^2) + 2.4 * sex - 0.048 * age+race+7.8$<br>$cAC=$ relaxed $Ac - (TR * 3.141/10)$<br>$cTHC=THc - (TH * 3.141/10)$<br>$cCC=Cc - (C * 3.141/10)$<br>$cCHC=CHc - (SS * 3.141/10)$<br>Sex: 1 for male, 0 for female<br>Race: –2 for Asians, 1.1 for African American, 0 for white or hispanic.<br>H in meters |
| | | Lee RC 2 [57] | $LM=0.244 * BW+7.8 * H+6.6 * sex - 0.098 * age+race - 3.3$<br>Sex: 1 for male, 0 for female<br>Race: –1.2 for Asians, 1.4 for African America, 0 for white of Hispanic.<br>H in meters |
| | | Doupe [60] | $LM= (H * (0.031 * cTHC^2 + 0.064 * cCC^2 + 0.089 * cAC^2) - 3.006)$<br>$cTHC=THc - (TH * 3.141/10)$<br>$cCC=Cc - (C * 3.141/10)$<br>$cAC=$ relaxed $Ac - (TR * 3.141/10)$ |
| | | ISAK 5 components model [48,49] | $LM= [((ZMUS * 5.4) + 24.5)/ (170.18/H)^3]$<br>$\sum 5 Circumferences= (cAC+FAc+cTHC+cCC+cCHC)$<br>$ZMUS = ((\sum 5 Circumferences*(170.18/ H) - 207.21))/ 13.74$<br>Corrections:<br>$cAC=$ relaxed $Ac - (TR * 3.141/10)$<br>$cTHC=THc - (TH * 3.141/10)$<br>$cCC=Cc - (C * 3.141/10)$<br>$cCHC=CHc - (SS * 3.141/10)$ |
| | | Janmahasatian [61] | $LM$ men$= (9270 * BW)/ (8780 + (244 * BMI))$<br>$LM$ women$= (9270 * BW)/ (6680 + (216 * BMI)$ |
| | | Olshvang [62] | $LM$ men $=19.363 + (0.001 * age) + (0.064 * H) + (0.756 * BW) -(0.366*WC) +$ or – race<br>+0.231 for Hispanics<br>+0.432 for African descendants<br>–1.007 for others<br>$LM$ women $=-10.683 - (0.039 * age) + (0.186 * H) + (0.383 * BW) - (0.043*WC) +$ or – race<br>–0.059 for Hispanics<br>+1.085 for African descendants<br>–0.34 for others |
| | | Chien [63] | $LM= 27.479 + (0.726 * BW) - (3.383* sex) - (0.672 * BMI) + (0.514 * Fac) - (0.245* HC)$<br>1 for male<br>0 for female |

*(Continued)*

| Variable | Definition | Author | Equation |
|---|---|---|---|
| | | Lee DH [50] | LM men = −1.401 − (0.010 * age) + (0.100 * H) + (0.632 * BW) − (0.225 * WC) + (0.315 * Ac) + (0.091 * Cc) + (0.040 * THc) − (0.304 * TR) − (0.021 * SS) + race<br>0.120 for Mexican<br>0.097 for Hispanic<br>0.463 for Afro descendants<br>− 0.661 for others<br>LM women = −9.193 − (0.045 * age) + (0.158 * H) + (0.410 * BW) − (0.040 * WC) + (0.095 * Ac) + (0.193 * Cc) − (0.105 * THc) − (0.152 * TR) − (0.004 * SS) + race<br>−0.306 for Mexican<br>0.082 for Hispanic<br>1.235 for Afro descendants<br>− 0.196 for others |
| | | Kulkarni equation 3 [64] | LM men = 13.78 − (0.018 * age) + (0.064 * H) + (0.697 * BW) − (5.842 * log ∑4SK)<br>LM women = 1.689 − (0.014 * age) + (0.120 * H) + (0.499 * BW) − (3.315 * log ∑4SK)<br>∑4SK = BI + TR + SS + SI |
| | | Kulkarni equation 4 [64] | LM men = 10.385 − (0.005 * age) + (0.103 * H) + (0.680 * BW) + (0.288 * Ac) + (0.130 * Cc) − (0.183 * HC) − (5.278 * log ∑4SK)<br>LM women = 10.632 − (0.009 * age) + (0.102 * H) + (0.592 * BW) + (0.055 * Ac) + (0.043 * Cc) − (0.158 * HC) − (3.174 * log ∑4SK)<br>∑4SK = BI + TR + SS + SI |
| | | Salamat [56] | LM = 14.966 + (0.588 * BMI) + (18.694 * sex) + (0.137 * WC) − (0.138 * age)<br>0 for female<br>1 for men |

BD: Body density

FP (%): Fat percent

BMI (Kg/m²): Body mass index

H (cm): height

BW (Kg): Body weight

FM (Kg): Fat mass

LP (%): Lean percentage

LM (Kg): Lean mass

ZMUS: Z muscular score

BM (Kg): bone mass

RM (Kg): residual mass

LM (Kg): lean mass

Skinfolds (SK): BI, bicipital; TR, tricipital; SS, subscapular; SI, supraliiac; TH, thigh; C, calf; SSP, supraspinal;

Circumferences: Ac, arm circumference; FAc, forearm circumference; THc, thigh circumference; Cc, calf circumference; CHc, chest circumference; HC, hip circumference; WC, waist circumference;

cAC: arm circumference corrected for triceps skinfold

cTHC: thigh circumference corrected for front thing skinfold

cCC: calf circumference corrected for medial calf skinfold

cCHC: chest circumference corrected for subscapular skinfold

Diameters (cm): WD, wrist diameter; FD, femur diameter

developed for able-bodied individuals [65]. Their inclusion was also justified by their practical applicability in field settings, as they rely on accessible and straightforward anthropometric variables such as body weight, height, skinfolds thickness, and circumferences. Furthermore, the selected equations are relatively simple to apply, which enhances their feasibility in routine assessments.

## Statistical analysis

Statistical analyses were conducted using the SPSS statistical software. Descriptive analyses were performed to summarize the continuous quantitative variables using means and standard deviations. The Shapiro-Wilk test was used to determine whether the data distribution was parametric or nonparametric. Most of the variables followed a normal distribution, except the results for LP and LM from DEXA, FP from Deuremberg, Gomez Ambrosi, Lean (with BMI) and Lee DH equations, FM from Lee DH and Salamat equations, LP from Lee RC equation and LM from Lee RC 2, Lee DH, Kulkarni equation 4 and Salamat equations. Therefore, a paired t-test was conducted to compare FP estimates from the equations with those obtained by DEXA, except for the equations Deuremberg, Gómez-Abrosi, Lean (using BMI) and Lee DH, which showed non-normally distributed data. For these cases, as well as well as for the comparison of FM from equations Lee DH and Salamat (also non-normally distributed), the Wilcoxon signed-rank test was used. The Wilcoxon test was also applied to compare LP and LM estimates from all equations with the corresponding DEXA measurements. Given the number of comparisons, we applied the Bonferroni correction [70] to control for multiple testing. For FP, where 34 equations were tested, the adjusted significance level was set at $p < 0.0015$. For FM, with 6 comparisons, the threshold was $p < 0.008$; for LP, with 3 comparisons, $p < 0.017$ and for LM, with 13 comparisons, $p < 0.004$.

The mean bias, defines as the average of the differences between the results estimated by each anthropometric equation and that obtained by DEXA, was calculated to assess systematic measurement error. In addition, the standard error of the difference (SED) was computed to quantify the precision of the estimated bias, providing an indicator of the uncertainty around the mean difference and confidence intervals were also calculated at the 95% level. To further evaluate the agreement between methods, the differences between each equation and the DEXA results were analyzed using one-sample t-test to identify equations whose estimates were not significantly different from those of DEXA. Absolute agreement was then assessed using Bland-Altman method, and 95% limits of agreement (LoA) were calculated. The relationship between the estimates measured by DEXA (dependent variable) and the results estimated by each equation was also evaluated using Reduced Major Axis (RMA) regression. This approach accounts for measurement error in both variables and provides slope and intercept, which help identify systematic or proportional bias. In this method, ideal agreement is indicated by an intercept of 0 and a slope of 1 [71].

To assess the concurrent validity of each equation with the DEXA, the Pearson Correlation Coefficient was calculated. The Spearman Correlation Coefficient was used to explore the correlation between the DEXA results and the equations if the data did not follow a normal distribution. A Pearson or Spearman correlation coefficient larger than 0.7 indicates very strong relationship; between 0.4 and 0.69, a strong relationship; between 0.3 and 0.39, a moderate relationship; between 0.2 and 0.29, a weak relationship; and between 0.01 and 0.19, no relationship. The intraclass correlation coefficient (ICC) was calculated using two distinct approaches. The first approach focused on estimating the ICC for absolute agreement and assessing whether the results provided by each equation precisely matched those obtained using DEXA. The second approach involved estimating the ICC for consistency, which evaluated the relative order of measurements between the results produced by the equations and those from DEXA to verify whether the measurements maintained the same relationship, even if they differed in absolute magnitude. An ICC value of 0 indicated poor agreement between instruments, values between 0.01–0.20 indicated slight agreement, 0.21–0.4 indicated fair agreement, 0.41–0.60 indicated moderate agreement, 0.61–0.80 indicated substantial agreement and 0.81–1.00 indicated almost perfect agreement.

## Results

Twenty-seven athletes were included, of whom 22 were men (81.4%) and 5 were women (18.5%) with a mean age of 32±7.4 years. Six athletes belong to the para-athletics team, 3 to para-cycling, 4 to para-powerlifting, 1 to wheelchair fencing, 2 to paralympic swimming, 2 to wheelchair tennis, 3 to para-triathlon, and 4 to sitting volleyball. Regarding the type of amputation, three athletes (10.71%) had hip disarticulation, nine (33.3%) had transfemoral amputation, and 15 (55.5%) had transtibial amputation. Table 2 summarizes the general characteristics of the sample, including age, body weight, height, BMI, anthropometric measurements such as circumference and skin folds, and DEXA results for FP, FM, LP, and LM.

Table 3 shows DEXA measurements and the results for each equation with their standard deviation. Table 4 shows the differences in the results between each equation and DEXA, the standard error of the difference, the ICC for absolute agreements and consistency and the Pearson or Spearman correlation coefficients. This table also shows the slope and the intercept of the RMA regression and the LoA from the Bland-Altman analysis for each equation.

### Concurrent validity and reliability of fat percentage equations

Regarding the concurrent validity of prediction equations for FP compared to DEXA in athletes with unilateral lower-limb amputation, we found that the equations showing very strong correlation with DEXA were Durnin and Womersley with Siri (r=0.82), Durnin and Womersley with Brozek (r=0.84), Sloan with Siri (r=0.78), Sloan with Brozek (r=0.78), Whithers with Siri (r=0.75), Whithers with Brozek (r=0,75), White with Siri (r=0.72), White with Brozek (r=0.72), Eston (r=0.76), Evans (r=0.84), O´Connor (r=0.76), Carter (r=0.82), Faulkner (r=0.73), Yuhasz (r=0.84), Hastuti (r=0.75), Lean log

Table 2. General characteristics of the sample, including age, body weight, height, BMI, anthropometric measurements (circumference and skin folds), and DEXA results for FP, FM, LP, and LM.

| Variable | Mean±SD (n=27) |
|---|---|
| Age (years) | 32.85±7.38 |
| Body weight (Kg) | 69.52±13.39 |
| Height (cm) | 167.63±6.8 |
| BMI (Kg/m²) | 24.6±3.68 |
| DEXA fat percentage (%) | 27.1±6.83 |
| DEXA fat mass (Kg) | 18.32±6.1 |
| DEXA lean percentage (%) | 50.66±9.24 |
| DEXA lean mass (Kg) | 49.16±9.53 |
| DEXA bone mineral content (Kg) | 2.5±0.42 |
| Arm circumference (cm) | 32.05±4.54 |
| Thigh circumference (cm) | 55.59±4.88 |
| Calf circumference (cm) | 36.76±3.4 |
| Chest circumference (cm) | 100±8.6 |
| Waist circumference (cm) | 95±7.11 |
| Tricipital skinfold (mm) | 11.8±5.07 |
| Subscapular skinfold (mm) | 18.3±8.3 |
| Bicipital skinfold (mm) | 7±3.6 |
| Suprailiac skinfold (mm) | 23.07±8.07 |
| Supraspinal skinfold (mm) | 15.31±7.86 |
| Abdominal skinfold (mm) | 25.45±9.02 |
| Thigh skinfold (mm) | 14.4±6 |
| Calf skinfold (mm) | 9.77±5.26 |

**Table 3.** **Results from DEXA measurements and from the anthropometric equations estimating FP, FM, LP, and LM.**

| Fat percentage (%) | Mean | SD |
|---|---|---|
| **DEXA fat percentage** | **27.1** | **6.83** |
| Durnnin and Womersley [26] + Siri [46] | 24.47 | 6,54 |
| Durnnin and Womersley [26] + Brozek [47] | 23.85 | 6.04 |
| Forsyth and Sinning [20,27] + Siri [46] | 26.23 | 10.88 |
| Forsyth and Sinning [20,27] + Brozek [47] | 25.47 | 10.05 |
| Katch and MacArdle [28] + Siri [46] | 16.82 | 5.51 |
| Katch and MacArdle [28] + Brozek [47] | 16.79 | 5.09 |
| Nagamine and Suzuki [29] + Siri [46] | 18.66 | 6.14 |
| Nagamine and Suzuki [29] + Brozek [47] | 18.48 | 5.67 |
| Sloan [30] + Siri [46] | 16.55 | 7.02 |
| Sloan [30] + Brozek [47] | 16.53 | 6.48 |
| Wilmore and Behnke [31] + Siri [46] | 18.28 | 3.93 |
| Wilmore and Behnke [31] + Brozek [47] | 18.14 | 3.63 |
| Whithers [32,66] + Siri [46] | 19.33 | 6.13 |
| Whithers [32,66] + Brozek [47] | 19.1 | 5.66 |
| White [20,33] + Siri [46] | 14.01 | 4.08 |
| White [20,33] + Brozek [47] | 14.19 | 3.76 |
| Eston [34] | 18.35 | 4.51 |
| Evans [35] | 16.56 | 5.17 |
| O'Connor [36] | 18.35 | 4.51 |
| Carter [37] | 13.73 | 5.13 |
| Faulkner [38] | 18.25 | 5.06 |
| Slaughter [39] | 17.02 | 7.19 |
| Yuhasz [40] | 14.63 | 4.83 |
| Minematsu [41] | 22.48 | 4.92 |
| Hastuti [42] | 26.36 | 5.87 |
| Deurenberg [43] | 23.88 | 5.57 |
| Lean Equation 1, waist circumference [44] | 22.52 | 5.88 |
| Lean BMI [44] | 23.52 | 5.95 |
| Lean BMI and TR [44] | 22.10 | 6.45 |
| Lean log10 ∑4SK [44] | 24.31 | 6.23 |
| Gómez-Ambrosi [45] | 24.97 | 6.2 |
| ISAK 5 components model [48,49] | 26.74 | 5.42 |
| Lee DH [50] | 26.08 | 7.11 |
| Giro [51] | 21.43 | 4.13 |
| **Fat mass (Kg)** | **Mean** | **SD** |
| **DEXA fat mass** | **18.32** | **6.18** |
| Al-Guindan [52] | 17.61 | 5.8 |
| De Rose and Guimaraes [53] | 11.88 | 4.56 |
| ISAK 5 components model [48,49] | 19.03 | 5.33 |
| Heitmann [55] | 15.92 | 6.13 |
| Lee DH [50] | 17.28 | 5.79 |
| Salamat [56] | 21.98 | 6.27 |
| **Lean Percentage (%)** | **Mean** | **SD** |
| **DEXA lean percentage** | **50.66** | **9.24** |
| Lee RC [57] | 44.41 | 6.11 |

*(Continued)*

**Table 3.** (Continued)

| Fat percentage (%) | Mean | SD |
|---|---|---|
| Poortsman [58] | 46.38 | 6.25 |
| ISAK 5 components model [48,49] | 46.22 | 4.8 |
| **Lean Mass (Kg)** | **Mean** | **SD** |
| **DEXA lean mass** | **49.16** | **9.53** |
| De Rose and Guimaraes [53] | 40.47 | 6.62 |
| Heymsfield [53,59] | 32.5 | 10.76 |
| Lee RC 1 [57] | 36.51 | 6.6 |
| Lee RC 2 [57] | 28.94 | 5.18 |
| Doupe [60] | 38.44 | 8.84 |
| ISAK 5 components model [48,49] | 37.52 | 8.09 |
| Janmahasatian [61] | 44.49 | 5.19 |
| Olshvang [62] | 50.32 | 9.2 |
| Chien [63] | 47.79 | 8.3 |
| Lee DH [50] | 50.68 | 8.93 |

10 ∑4SK (r = 0.73) and ISAK 5 components model (r = 0.72). Additionally, equations that shows almost perfect agreement, based on the intraclass correlation coefficient with consistency approach (ICCc) were Durnin and Womersley with Siri (ICCc = 0.91; 95% CI: 0.81 to 0.96), Durnin and Womersley with Brozek (ICCc = 0.91; 95% CI: 0.81 to 0.96), Sloan with Siri (ICCc = 0.87; 95% CI: 0.72 to 0.94), Sloan with Brozek (ICCc = 0.83; 95% CI: 0.72 to 0.94), Whither with Siri (ICCc = 0.85; 95% CI: 0.67 to 0.93), Whither with Brozek (ICCc = 0.85; 95% CI: 0.66 to 0.93), Eston (ICCc = 0.82; 95% CI: 0.6 to 0.92), Evans (ICCc = 0.89; 95% CI: 0.77 to 0.95), O´Connor (ICCc = 0.82; 95% CI: 0.61 to 0.92), Carter (ICCc = 0.88; 95% CI: 0.74 to 0.95), Faulkner (ICCc = 0.82; 95% CI: 0.6 to 0.92), Yuhasz (ICCc = 0.88; 95% CI: 0.74 to 0.95), Hastuti (ICCc = 0.85; 95% CI: 0.68 to 0.93), Lean log 10 ∑4SK (ICCc = 0.87; 95% CI: 0.72 to 0.94), ISAK 5 components model (ICCc = 0.82; 95% CI: 0.61 to 0.92) and Lee DH (ICCc = 0.82; 95% CI: 0.6 to 0.92). However, due that some of these 95% confidence intervals were too wide, we consider that only the following equations produce substantial to almost perfect consistent rankings or relative measurements compared to DEXA: Durnin and Womersley with Siri, Durnin and Womersley with Brozek, Sloan with Siri, Sloan with Brozek, Evans, Carter, Yuhasz, Hastuti and Lean log 10 ∑4SK. On the other hand, when assessing absolute agreement between the equations and the DEXA results, only five equations exhibit an almost perfect level of agreement with DEXA, according to the Intraclass Correlation Coefficient with absolute agreement approach (ICCa): Durnin and Womersley with Siri (ICCa = 0.88; 95% CI: 0.61 to 0.95), Durnin and Womersley with Brozek (ICCa = 0.86; 95% CI: 0.4 to 0.95), Hastuti (ICCa = 0.86; 95% CI: 0.68 to 0.93), Lean log 10 ∑4SK (ICCa = 0.83; 95% CI: 0.54 to 0.93), ISAK 5 components model (ICCa = 0.83; 95% CI: 0.62 to 0.92) and Lee DH (ICCa = 0.85; 95% CI: 0.66 to 0.93). However, considering the 95% confidence intervals, only Durnin and Womersley with Siri, Hastuti, ISAK 5 components model, and Lee DH demonstrated substantial to almost perfect absolute agreement compared to the FP results provided by DEXA.

Even though the ICCc and ICCa values were almost perfect for and Durnin and Womersley with Brozek, their results statistically differ from the measurements provided with DEXA (3.21 ± 3.69% p < 0.001). On the other hand, we found that the results of the equations proposed by Forsyth and Sinning with Siri (mean difference 0.83 ± 9.16% p = 0.642) and Forsyth and Sinning with Brozek (mean difference 1.59 ± 8.52% p = 0.342) were not statistically different from DEXA, even though ICCc and ICCa showed only substantial correlations. This finding can be attributed to the high standard deviation of the differences, indicating substantial variability between Forsyth and Sinning equations and DEXA values across individuals. As a result, there is greater uncertainty in the estimation of the mean difference. This level of variability leads

**Table 4. Differences of the results between the equations and DEXA, as well as the ICCa and ICCc, Pearson correlation coefficient or de Spearman Correlation Coefficient, in case of non-normally distributed variables, RMA regression and the LoA of the Bland-Altman analysis.**

| Comparison DEXA fat percent vs | Mean difference ± standard deviation | Standard Error of the Difference | CI 95% of the difference | p value t-student or Wilcoxon | ICC (IC 95%) Absolute agreement | ICC (IC 95%) Consistency | Pearson or spearman correlation coefficients | RMA Regression | | Bland-Altman analysis | |
|---|---|---|---|---|---|---|---|---|---|---|---|
| | | | | | | | | Slope | Intercept | Lower LoA | Upper LoA |
| Fat percentage | | | | | | | | | | | |
| Durnin and Womersley [26] + Siri | 2.6±3.77 | 0.72 | 1.1; 4.08 | 0.001$^{ns}$ | 0.88 (0.61; 0.95) | 0.91 (0.81; 0.96) | 0.82 | 1.04 | 1.5 | -4.8 | 10.0 |
| Durnin and Womersley [26] + Brozek | 3.21±3.69 | 0.71 | 1.75; 4.67 | <0.001 | 0.86 (0.4; 0.95) | 0.91 (0.81; 0.96) | 0.84 | 1.13 | 0.1 | -4.0 | 10.4 |
| Forsyth and Sinning [20,27] + Siri | 0.83±9.16 | 1.77 | -2.79; 4.45 | 0.642$^{ns}$ | 0.67 (0.26; 0.85) | 0.66 (0.25; 0.85) | 0.55 | 0.63 | 10.6 | -17.1 | 18.8 |
| Forsyth and Sinning [20,27] + Brozek | 1.59±8.52 | 1.64 | -1.78; 4.96 | 0.342$^{ns}$ | 0.67 (0.29; 0.85) | 0.67 (0.29; 0.85) | 0.55 | 0.68 | 9.8 | -15.1 | 18.3 |
| Katch and MacArdle [28] + Siri | 10.24±5.18 | 0.99 | 8.19; 12.29 | <0.001 | 0.43 (-0.19; 0.78) | 0.79 (0.54; 0.90) | 0.66 | 1.24 | 6.3 | 0.1 | 20.4 |
| Katch and MacArdle [28] + Brozek | 10.27±5.12 | 0.99 | 8.25; 12.3 | <0.001 | 0.41 (-0.19; 0.77) | 0.78 (0.52; 0.9) | 0.66 | 1.34 | 4.6 | 0.2 | 20.3 |
| Nagamine and Suzuki [29] + Siri | 8.4±5.25 | 1.01 | 6.32; 10.48 | <0.001 | 0.54 (-0.23; 0.84) | 0.80 (0.57; 0.91) | 0.68 | 1.11 | 6.3 | -1.9 | 18.7 |
| Nagamine and Suzuki [29] + Brozek | 8.58±5.14 | 0.99 | 6.54; 10.61 | <0.001 | 0.51 (-0.23; 0.83) | 0.8 (0.56; 0.91) | 0.68 | 1.20 | 4.8 | -1.5 | 18.7 |
| Sloan [30] + Siri | 10.51±4.64 | 0.89 | 8.67; 12.34 | <0.001 | 0.53 (-0.16; 0.85) | 0.87 (0.72; 0.94) | 0.78 | 0.97 | 11.0 | 1.4 | 19.6 |
| Sloan [30] + Brozek | 10.52±4.47 | 0.86 | 8.76; 12.29 | <0.001 | 0.51 (-0.15; 0.84) | 0.83 (0.72; 0.94) | 0.78 | 1.05 | 9.7 | 1.8 | 19.3 |
| Wilmore and Behnke [31] + Siri | 8.77±5.35 | 1.03 | 6.66; 10.89 | <0.001 | 0.39 (-0.22; 0.74) | 0.7 (0.34; 0.86) | 0.62 | 1.74 | -4.7 | -1.7 | 19.3 |
| Wilmore and Behnke [31] + Brozek | 8.92±5.37 | 1.03 | 6.80; 11.05 | <0.001 | 0.37 (-0.22; 0.72) | 0.68 (0.3; 0.86) | 0.62 | 1.88 | -7.0 | -1.6 | 19.4 |
| Whithers [32,66] + Siri | 7.73±4.67 | 0.90 | 5.89; 9.58 | <0.001 | 0.61 (-0.23; 0.87) | 0.85 (6.7; 0.93) | 0.75 | 1.11 | 5.6 | -1.4 | 16.9 |
| Whithers [32,66] + Brozek | 7.97±4.58 | 0.88 | 6.15; 9.78 | <0.001 | 0.58 (-0.22; 0.86) | 0.85 (0.66; 0.93) | 0.75 | 1.21 | 4.1 | -1.0 | 16.9 |
| White [20,33] + Siri | 13.04±4.84 | 0.93 | 11.13; 14.96 | <0.001 | 0.29 (-0.12; 0.68) | 0.77 (0.5; 0.9) | 0.72 | 1.67 | 3.6 | 3.6 | 22.5 |
| White [20,33] + Brozek | 12.87±4.9 | 0.94 | 10.93; 14.80 | <0.001 | 0.28 (-0.12; 0.66) | 0.75 (0.5; 0.89) | 0.72 | 1.82 | 1.3 | 3.3 | 22.5 |
| Eston [34] | 8.71±4.5 | 0.86 | 6.93; 10.49 | <0.001 | 0.5 (-0.2; 0.82) | 0.82 (0.6; 0.92) | 0.76 | 1.51 | -0.7 | -0.1 | 17.5 |
| Evans [35] | 10.5±3.73 | 0.72 | 9.02; 11.97 | <0.001 | 0.5 (-0.12; 0.83) | 0.89 (0.77; 0.95) | 0.84 | 1.32 | 5.2 | 3.2 | 17.8 |
| O'Connor [36] | 8.71±4.5 | 0.86 | 6.93; 10.49 | <0.001 | 0.5 (-0.2; 0.82) | 0.82 (0.61; 0.92) | 0.76 | 1.51 | -0.7 | -0.1 | 17.5 |
| Carter [37] | 13.33±3.93 | 0.76 | 11.78; 14.88 | <0.001 | 0.37 (-0.08; 0.76) | 0.88 (0.74; 0.95) | 0.82 | 1.33 | 8.8 | 5.6 | 21.0 |
| Faulkner [38] | 8.81±4.67 | 0.90 | 6.96; 10.65 | <0.001 | 0.51 (-0.21; 0.83) | 0.82 (0.6; 0.92) | 0.73 | 1.35 | 2.5 | -0.3 | 18.0 |
| Slaughter [39] | 10.04±5.95 | 1.14 | 7.68; 12.4 | <0.001 | 0.48 (-0.23; 0.81) | 0.78 (0.52; 09) | 0.64 | 0.95 | 10.9 | -1.6 | 21.7 |
| Yuhasz [40] | 12.43±3.82 | 0.74 | 10.91; 13.94 | <0.001 | 0.4 (-0.087; 0.78) | 0.88 (0.74; 0.95) | 0.84 | 1.41 | 6.4 | 4.9 | 19.9 |
| Minematsu [41] | 4.58±6.13 | 1.18 | 2.15; 7 | <0.001 | 0.54 (-0.03; 0.79) | 0.64 (0.21; 0.84) | 0.5 | 1.39 | -4.1 | -7.4 | 16.6 |

*(Continued)*

**Table 4.** (Continued)

| Comparison DEXA fat percent vs | Mean difference ± standard deviation | Standard Error of the Difference | CI 95% of the difference | p value t-student or Wilcoxon | ICC (IC 95%) Absolute agreement | ICC (IC 95%) Consistency | Pearson or spearman correlation coefficients | RMA Regression | | Bland-Altman analysis | |
|---|---|---|---|---|---|---|---|---|---|---|---|
| | | | | | | | | Slope | Intercept | Lower LoA | Upper LoA |
| Hastuti [72] | 0.7±4.55 | 0.88 | −1.1; 2.5 | 0.434ns | 0.86 (0.68; 0.93) | 0.85 (0.68; 0.93) | 0.75 | 1.16 | −3.6 | −8.2 | 9.6 |
| Deurenberg [43] | 3.18±7.34 | 1.41 | 0.27; 6.1 | 0.033*ns | 0.43 (−0.14; 0.73) | 0.47 (−0.17; 0.76) | 0.25** | 1.23 | −2.2 | −11.2 | 17.6 |
| Lean Equation 1, waist circumference [44] | 4.54±5.66 | 1.07 | 2.34; 6.74 | <0.001 | 0.67 (0.08; 0.86) | 0.76 (0.48; 0.89) | 0.63 | 1.16 | 0.9 | −6.6 | 15.6 |
| Lean BMI [44] | 3.54±7.55 | 1.45 | 0.55; 6.52 | 0.007*ns | 0.43 (−0.15; 0.73) | 0.47 (−0.17; 0.76) | 0.28** | 1.15 | 0.1 | −11.3 | 18.3 |
| Lean BMI and TR [44] | 4.96±6.37 | 1.23 | 2.44; 7.48 | <0.001 | 0.6 (0.026; 0.83) | 0.7 (0.34; 0.86) | 0.54 | 1.06 | 3.7 | −7.5 | 17.4 |
| Lean log10 ∑4SK [44] | 2.75±4.4 | 0.85 | 1.01; 4.5 | 0.003ns | 0.83 (0.54; 0.93) | 0.87 (0.72; 0.94) | 0.78 | 1.10 | 0.4 | −5.9 | 11.4 |
| Gómez-Ambrosi [45] | 2.09±7.43 | 1.43 | −0.85; 5.03 | 0.058*ns | 0.51 (−0.04; 0.77) | 0.52 (−0.054; 0.78) | 0.28** | 1.10 | −0.4 | −12.5 | 16.7 |
| ISAK 5 components model [48,49] | 0.32±4.8 | 0.92 | −1.58; 2.22 | 0.733ns | 0.83 (0.62; 0.92) | 0.82 (0.61; 0.92) | 0.72 | 1.26 | −6.6 | −9.1 | 9.7 |
| Lee DH [50] | 0.98±5.49 | 1.06 | −1.19; 3.15 | 0.532*ns | 0.82 (0.6; 0.92) | 0.82 (0.6; 0.92) | 0.64** | 0.96 | 2.0 | −9.8 | 11.7 |
| Giro [51] | 5.63±5.03 | 0.97 | 0.97; 3.64 | 0.000 | 0.58 (−0.17; 0.84) | 0.75 (0.46; 0.89) | 0.68 | 1.65 | −8.3 | −4.2 | 15.5 |
| **Fat mass** | | | | | | | | | | | |
| Al-Guindan [52] | 0.71±4.63 | 0.89 | −1.12; 2.55 | 0.431ns | 0.83 (0.62; 0.92) | 0.82 (0.62; 0.92) | 0.68 | 1.07 | −0.4 | −8.4 | 9.8 |
| De Rose and Guimaraes [53] | 6.45±3.39 | 0.65 | 5.11; 7.79 | <0.001 | 0.64 (−0.2; 0.89) | 0.89 (0.76; 0.95) | 0.85 | 1.36 | 2.2 | −0.2 | 13.1 |
| 5 components ISAK [48,49] | −0.71±3.64 | 0.70 | −2.15; 0.73 | 0.323ns | 0.89 (0.76; 0.95) | 0.89 (0.76; 0.95) | 0.81 | 1.16 | −3.7 | −7.8 | 6.4 |
| Heitmann [55] | 2.4±5.44 | 1.05 | 0.25; 4.55 | 0.030ns | 0.73 (0.4; 0.88) | 0.75 (0.47; 0.89) | 0.59 | 1.01 | 2.3 | −8.3 | 13.1 |
| Lee DH [50] | 1.04±4.35 | 0.84 | −0.68; 2.76 | 0.280*ns | 0.85 (0.66; 0.93) | 0.85 (0.67; 0.93) | 0.7** | 1.07 | −0.1 | −7.5 | 9.6 |
| Salamat [56] | −3.65±6.35 | 1.22 | −6.17; −1.14 | 0.002* | 0.59 (0.11; 0.81) | 0.65 (0.23; 0.84) | 0.6** | 0.99 | −3.3 | −16.1 | 8.8 |
| **Lean percentage** | | | | | | | | | | | |
| Lee RC [57] | 6.25±11.87 | 2.29 | 1.55; 10.94 | 0.015* | −0.27 (−1.26; 0.36) | −0.35 (−1.97; 0.38) | −0.13** | −1.51 | 117.8 | −17.0 | 29.5 |
| Poortsman [58] | 4.28±10.94 | 2.11 | −0.05; 8.61 | 0.039*ns | 0.07 (−0.83; 0.55) | 0.073 (−1.04; 0.58) | 0.12** | 1.48 | −17.9 | −17.2 | 25.7 |
| 5 component ISAK [48,49] | 4.44±9.03 | 1.74 | 0.86; 8 | 0.020*ns | 0.35 (−0.26; 0.69) | 0.4 (−0.32; 0.73) | 0.22** | 1.93 | −38.3 | −13.3 | 22.1 |
| **Lean mass** | | | | | | | | | | | |
| De Rose and Guimaraes [53] | 8.69±4,15 | 0.80 | 7.05; 10.33 | <0.001* | 0.72 (−0.17; 0.92) | 0.93 (0.85; 0.97) | 0.87** | 1.44 | −9.1 | 0.6 | 16.8 |
| Heymsfield [53,59] | 16.66±9.15 | 1.76 | 13.04; 20.28 | <0.001* | 0.41 (−0.21; 0.76) | 0.75 (0.44; 0.88) | 0.58** | 0.89 | 20.4 | −1.3 | 34.6 |
| Lee RC 1 [57] | 12.64±5.26 | 1.01 | 10.56; 14.72 | <0.001* | 0.53 (−0.15; 0.85) | 0.89 (0.75; 0.95) | 0.79** | 1.44 | −3.6 | 2.3 | 22.9 |
| Lee RC 2 [57] | 20.22±5.62 | 1.08 | 18; 22.44 | <0.001* | 0.28 (−0.07; 0.68) | 0.85 (0.66; 0.93) | 0.78** | 1.84 | −4.1 | 9.2 | 31.2 |
| Douple [60] | 10.71±6.01 | 1.16 | 8.34; 13.1 | <0.001* | 0.64 (−0.21; 0.89) | 0.88 (0.74; 0.95) | 0.78** | 1.08 | 7.7 | −1.1 | 22.5 |
| ISAK 5 components model [48,49] | 11.64±4.42 | 0.85 | 9.89; 13.38 | <0.001* | 0.64 (−0.12; 0.9) | 0.93 (0.85; 0.97) | 0.85** | 1.18 | 5.0 | 3.0 | 20.3 |
| Janmahasatian [61] | 4.67±8.87 | 1.71 | 1.16; 8.18 | 0.025ns | 0.44 (−11; 0.74) | 0.5 (−0.10; 0.77) | 0.48** | 1.84 | −32.5 | −12.7 | 22.1 |
| Olshvang [62] | −1.16±5.49 | 1.06 | −3.33; 1.01 | 0.058*ns | 0.91 (0.8; 0.96) | 0.91 (0.79; 0.96) | 0.76** | 1.04 | −3.0 | −11.9 | 9.6 |

*(Continued)*

**Table 4.** (Continued)

| Comparison DEXA fat percent vs | Mean difference ± standard deviation | Standard Error of the Difference | CI 95% of the difference | p value t-student or Wilcoxon | ICC (IC 95%) Absolute agreement | ICC (IC 95%) Consistency | Pearson or spearman correlation coefficients | RMA Regression | | Bland-Altman analysis | |
|---|---|---|---|---|---|---|---|---|---|---|---|
| | | | | | | | | Slope | Intercept | Lower LoA | Upper LoA |
| Chien [63] | 1.37±6.04 | 1.16 | −1.02; 3.76 | 0.755*ns | 0.87 (0.72; 0.94) | 0.87 (0.72; 0.94) | 0.74** | 1.15 | −5.7 | −10.5 | 13.2 |
| Lee DH [50] | −1.52±4.11 | 0.79 | −3.15; 0.11 | 0.008*ns | 0.94 (0.87; 0.97) | 0.95 (0.89; 0.98) | 0.84** | 1.07 | −4.9 | −9.6 | 6.5 |
| Kulkarni equation 3 [64] | 0.44±5.9 | 1.14 | −1.89; 2.78 | 0.683*ns | 0.84 (0.65; 0.93) | 0.84 (0.64; 0.93) | 0.82** | 1.65 | −31.4 | −11.1 | 12.0 |
| Kulkarni equation 4 [64] | −11.41±4.65 | 0.90 | −13.25; −9.57 | 0.000* | 0.71 (−0.14; 0.93) | 0.95 (0.88; 0.96) | 0.83** | 0.88 | −4.4 | −20.5 | −2.3 |
| Salamat [56] | −3.08±5.2 | 1 | −5.13; −1.02 | 0.003* | 0.89 (0.69; 0.96) | 0.91 (0.81; 0.96) | 0.71** | 1.06 | −6.4 | −13.3 | 7.1 |

*p value from Wilcoxon test

**Spearman correlation coefficient

ns non-significant result

RMA reduced major axis regression

LoA limits of agreement

to a larger standard error of the difference (SED) (1.77 and 1.64 for Forsyth and Sinning with Siri and Brozek respectively), hence, the paired t-test did not detect statistically significant difference from zero. Consequently, the Durning and Womersley equations exhibited lower standard deviations of the difference, resulting in lower standard errors of the mean difference. Although the paired t-test indicated statistically significant difference, the relatively small standard error suggest that the estimates of the mean differences are precise. Four equations that showed almost perfect agreement with DEXA and no statistically significant differences compared to DEXA with a relative low SED were: Hastuti (mean difference 0.7±4.55% p=0.434 SED=0.88), ISAK 5 components model (mean difference 0.32±4.8% p=0.733 SED=0.92), Durnin and Womersley with Siri (mean difference 2.6±3.77% p=0.0015 SED=0.72) and Lean log 10 ∑4SK (mean difference 2.75±4.4% p=0.003 SED=0.85). Nonetheless, despite the lack of statistical significance and relative low variability, the last two equations (Durnin and Womersley with Siri, and Lean log 10 ∑4SK) exhibited clinically meaningful biases. A bias exceeding 1% in FP estimation may be considered too large for practical applications in this population.

The equation Lee DH did not show statistically significant difference compared to DEXA measurements (mean difference 0.98±5.49% p=0.532 SED=1.06). However, the standard error of the mean difference exceeds 1%, indicating substantial uncertainty in the estimate of the mean bias. While the ICCc and ICCa showed significant agreement, the validation (Spearman) correlation coefficient was 0.64, suggesting only a moderate level of association between this equation and DEXA.

Regarding the Bland-Altman analysis, only six equations showed no statistically difference from DEXA: Forsyth and Sinning with Siri (Fig 1), Forsyth and Sinning with Brozek (Fig 2), Hastuti (Fig 3), Gomes-Ambrosi (Fig 4), the ISAK 5 components model (Fig 5) and Lee DH (Fig 6). All remaining equations exhibited either a large systematic bias or excessively wide LoA, indicating poor absolute agreement with DEXA and limiting their clinical utility. Concerning Forsyth and Sinning with Siri the mean bias was 0.83%, with a standard deviation of 9.16%. The 95% LoA ranged from −17.12 to 18.78%, indicating considerable variability between the two methods. Most data points fell below zero-difference line, suggesting that this equation tends to overestimate FP compared to DEXA. Similar results were observed for the Forsyth and Sinning

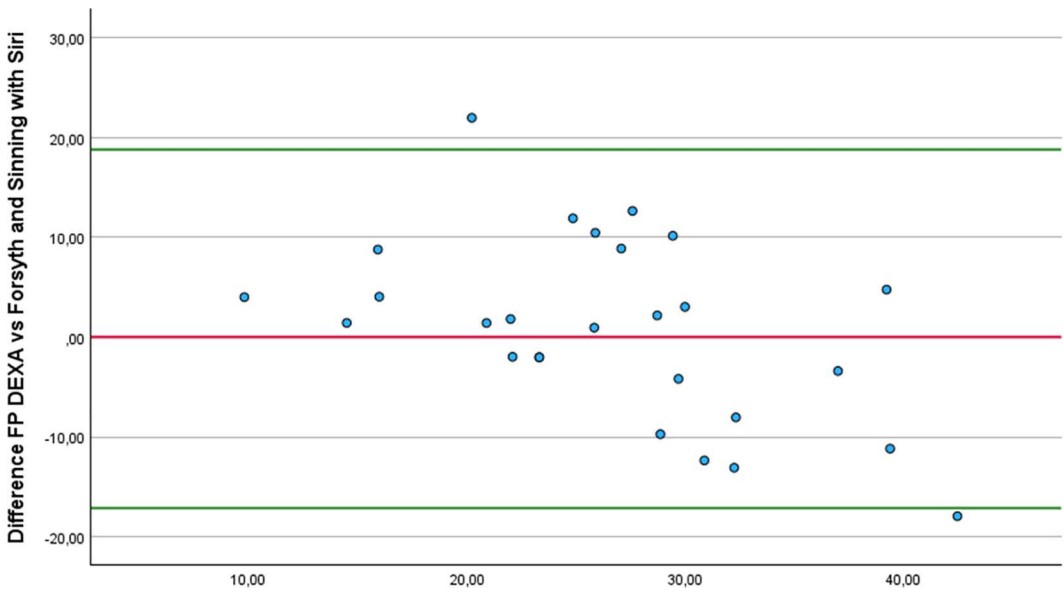

**Fig 1. Bland-Altman plot comparing FP estimates between DEXA and Forsyth and Sinning with Siri equation.**

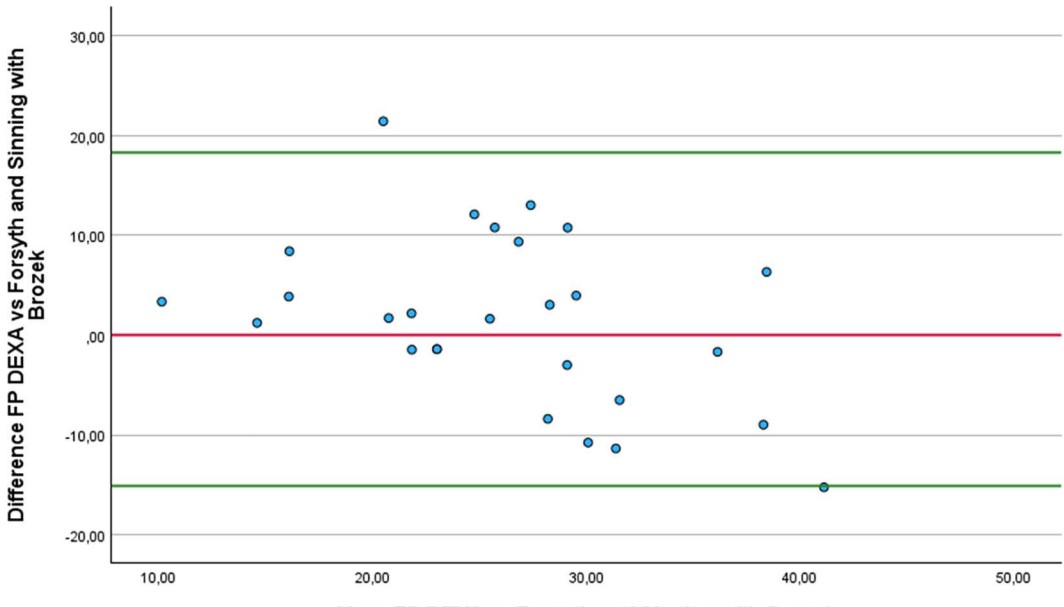

**Fig 2. Bland-Altman plot comparing FP estimates between DEXA and Forsyth and Sinning with Brozek equation.**

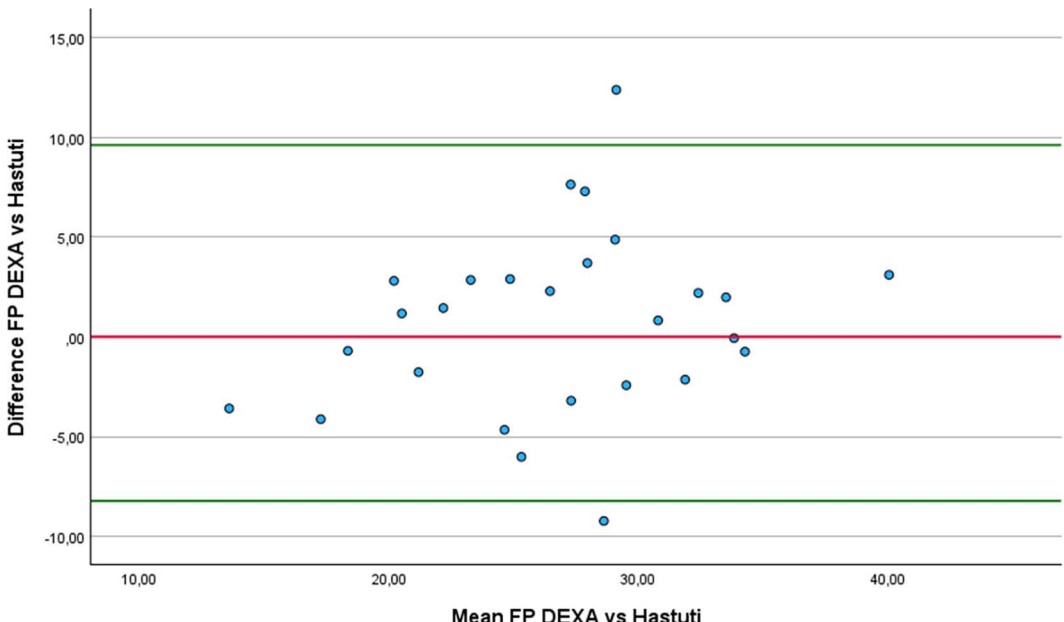

**Fig 3. Bland-Altman plot comparing FP estimates between DEXA and Hastuti equation.**

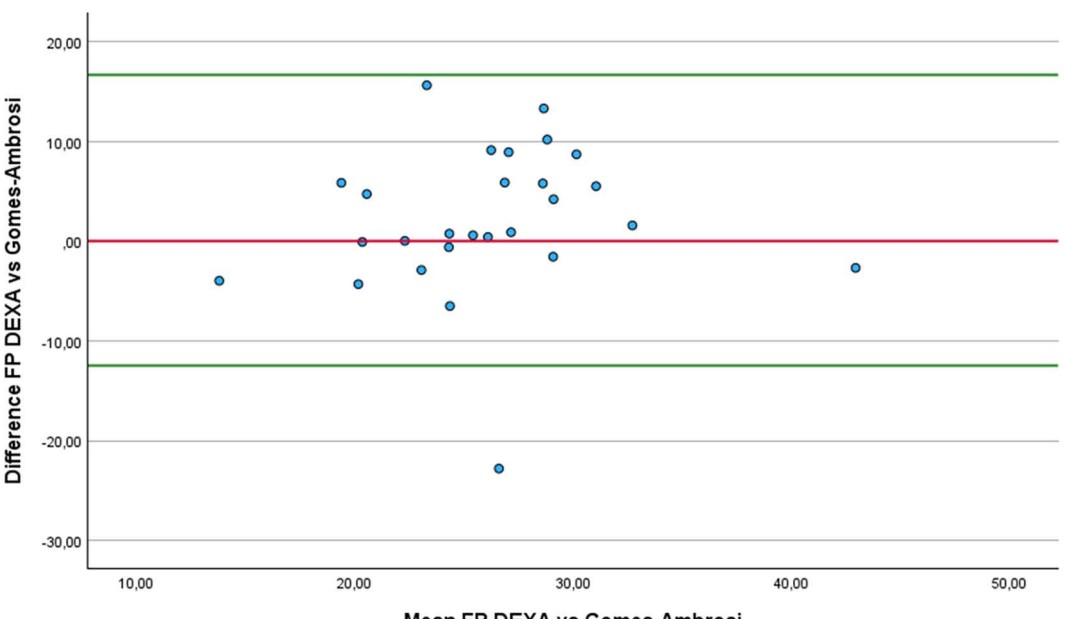

**Fig 4. Bland-Altman plot comparing FP estimates between DEXA and Gomes-Ambrosi equation.**

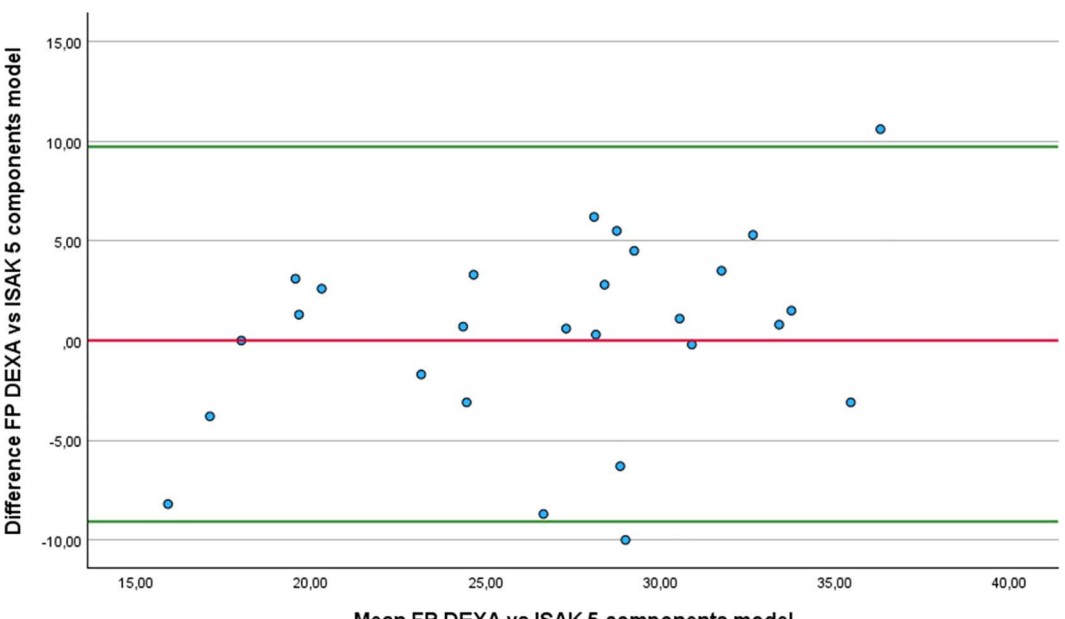

**Fig 5. Bland-Altman plot comparing FP estimates between DEXA and ISAK 5 components model equation.**

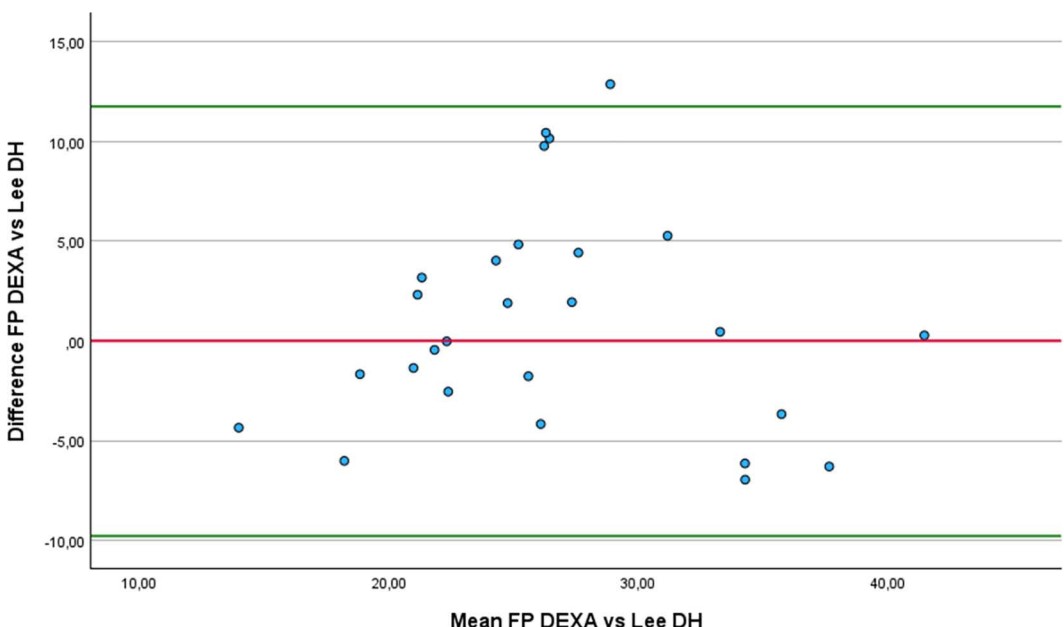

**Fig 6. Bland-Altman plot comparing FP estimates between DEXA and Lee DH equation.**

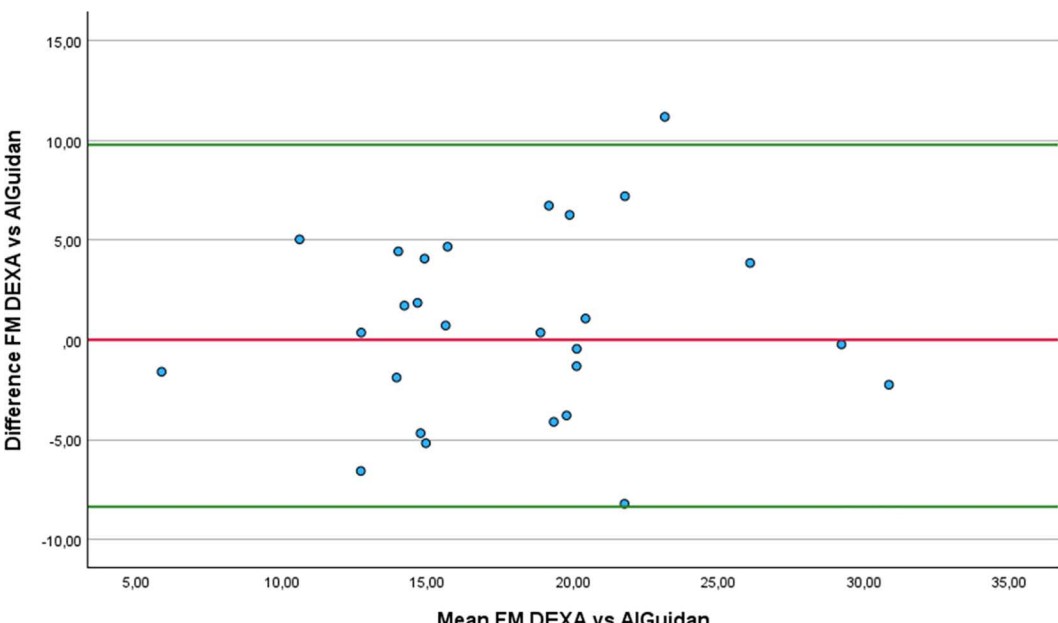

**Fig 7. Bland-Altman plot comparing FM estimates between DEXA and AIGuidan equation.**

with Brozek (mean bias of 1.59% and LoA ranging from −15.11 to 18.23%) and Gómes-Ambrosi (mean bias of 2.09% and LoA ranging from −12.47 to 16.65%). For these two equations, most data points were located above zero-difference line, indicating a trend to underestimate fat percentage compared to DEXA. Given the wide LoA, these differences are considered clinically unacceptable for use in this population. Therefore, the Forsyth and Sinning with Siri, Forsyth and Sinning with Brozek and the Gomes-Ambrosi do not demonstrate sufficient agreement with DEXA to be recommended for estimating FP in athletes with unilateral lower-limb amputation.

For the Hastuti, ISAK 5 components model, and Lee DH equations, a consistent trend of underestimation FP compared to DEXA was observed. The mean bias for each equation was below 1%, which indicates minimal systematic error. However, the LoA remained relatively wide. This implies that, when applying any of these equations, estimated FP may differ from DEXA measurement by as much as −8.22% to 9.62% for the Hastuti equation, −9.09% to 9.72% for the ISAK 5 components model, and −9.78 to 11. 74% for Lee DH equation, in 95% of the cases. Although the low bias is encouraging, the breadth of the LoA indicates substantial individual variability, limiting the clinical applicability if these equations for precise estimation of FP in athletes with unilateral lower-limb amputation.

Finally, regarding the results from the RMA regression, the four equations that demonstrated the best agreement with DEXA, considering mean difference, correlation coefficient, regression slope, and intercept, were: Hastuti, Lee DH, the ISAK 5 component model and Durning and Womersley with Siri. The Hastuti equation showed the most balanced performance, with minimal bias, high correlation, and regression parameters close to ideal. Lee DH also showed strong agreement, with a slope near 1 and low intercept, although its correlation was low. The Durning and Womersley with Siri equation showed a slope close to 1, but also a greater mean underestimation. The ISAK 5 component model presented a minimal average error but its elevated slope suggests potential overestimation of FP in individuals with high adiposity and underestimation in lean individuals.

Based on the analysis of validity and reliability of the equations, we recommend the Hastuti [42,72] and the ISAK 5 components model equations [48] to estimate FP in athletes with unilateral lower-limb amputation. The classic equations of Durning and Womersley with Siri could be also be considered, although the bias compared to DEXA is approximately 2.6%, which may limit its precision for this specific population.

## Concurrent validity and reliability of fat mass equations

For the FM equations, only 3 showed very strong correlation with DEXA: De Rose and Guimaraes (r = 0.85), ISAK 5 components model (r = 0.81) and Lee DH (R = 0.7), indicating good concurrent validity. Four equations tested in this study exhibit almost perfect relative agreement with DEXA: Al-Guindan (ICCc = 0.82; 95% CI: 0.62 to 0.92), De Rose and Guimaraes (ICCc = 0.89; 95% CI: 0.76 to 0.95), ISAK 5 components model (ICCc = 0.89; 95% CI: 0.76 to 0.95) and Lee DH (ICCc = 0.85; 95% CI: 0.67 to 0.93). However, only Al-Guindan (ICCa = 0.83; 95% CI: 0.62 to 0.92), ISAK 5 components model (ICCa = 0.89; 95% CI: 0.76 to 0.95) and Lee DH (ICCa = 0.85; 95% CI: 0.66 to 0.93) showed almost perfect absolute agreement compared to DEXA with no statistically significant differences to this evaluation method (mean difference 0.71 ± 4.63% p = 0.431 SED = 0.89; mean difference −0.71 ± 3.64% p = 0.323 SED = 0.7; mean difference 1.04 ± 4.35% p = 0.280 SED = 0.84 respectively). Also, their SED were below to 1 Kg.

Regarding FM estimation, four equations did not show significant differences with DEXA: the Al-Guindan (Fig 7), ISAK 5 components model (Fig 8), Lee DH (Fig 9) and Heitman equations. However, the bias of Heitman equation compared to DEXA results was clinically high (2.4%). The Al-Guindan equation underestimated FM by an average of 0.71 Kg, and with 95% LoA ranging from −8.36 to 9.78 Kg. Similarly, the Lee DH equation showed a mean underestimation of 1.04 Kg, with LoA between −7.49 and 9.67 Kg. These wide LoA values suggest substantial variability in individual estimates, limiting the clinical usefulness of both equations for accurately assessing FM in athletes with lower-limb amputation. In contrast the ISAK 5 components model slightly overestimate FM 0.71 Kg, with comparatively narrower LoA (−7.84 to 6.42 Kg).

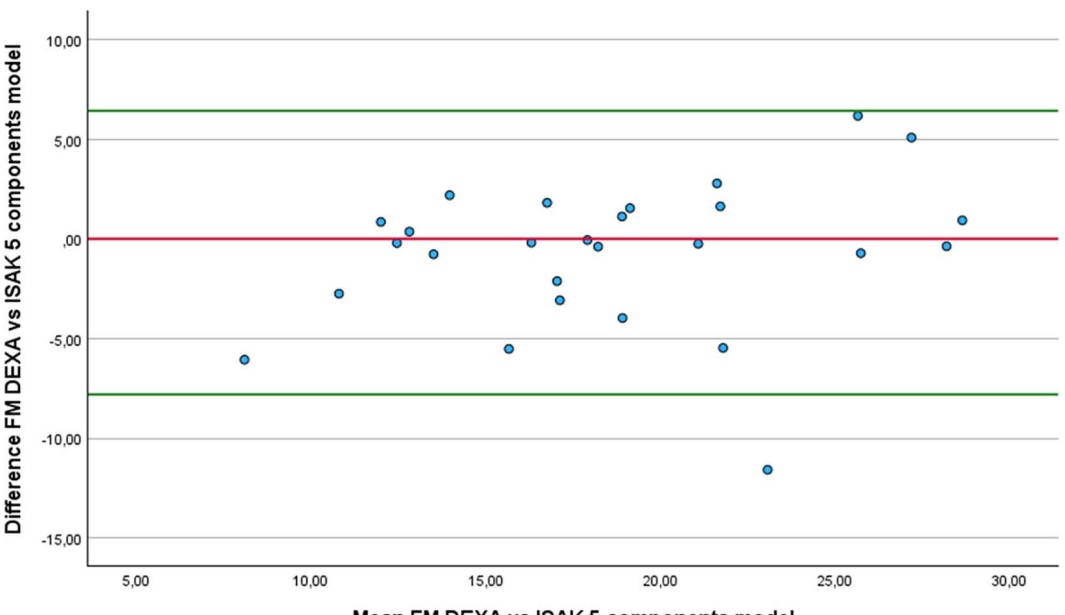

**Fig 8. Bland-Altman plot comparing FM estimates between DEXA and ISAK 5 components model equation.**

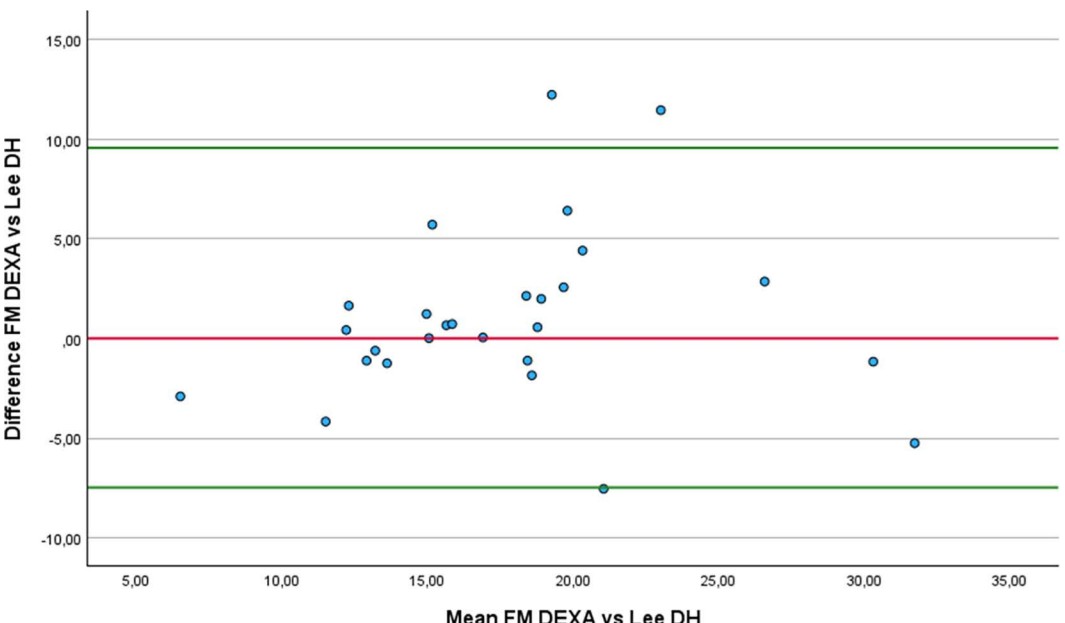

**Fig 9. Bland-Altman plot comparing FM estimates between DEXA and Lee DH equation.**

Although this equation also exhibits considerable variation, the reduced range of disagreement indicates relatively better agreement with DEXA.

In the comparison of the anthropometric equations with the DEXA, using the RMA regression, notable differences were observed in terms of accuracy and agreement. The ISAK 5 component model demonstrated the smallest mean difference with DEXA (0.71 Kg), high correlation (r = 0.81), and an acceptable slope (1.16), suggesting good overall performance with slight overestimation at higher FM levels. The Al-Guidan and Lee DH also showed modest difference from DEXA (−0.71 Kg and −1.04 Kg, respectively) and slopes close to 1 (1.07 for both), though with moderate correlations (r = 0.68 and 0.7), indicating acceptable predictive capacity. In contrast, the De Rose and Guimaraes equation significantly underestimates FM (−6.44 Kg), despite a high correlation (r = 0.85), and had an elevated slope (1.36), suggesting overestimation at higher FM levels. The Heitmann equation showed a low correlation (r = 0.59) and moderate underestimation (−2.4 Kg), while Salamat equation resulted in considerable overestimation with low correlation (r = 0.6). Therefore, we recommend, in the first place, the ISAK 5 components model, followed by Lee HD equations, to estimate FM in athletes with unilateral lower-limb amputation.

## Concurrent validity and reliability of lean percentage

All equations used to estimate LP in athletes with unilateral lower-limb amputation demonstrated weak or limited absolute and relative agreement with DEXA, as well as low correlation coefficients. Although the equations from Poortsman and ISAK 5 components model did not show statistically significant differences compared to DEXA, their large biases and wide standard deviations of the difference indicate that these equations lack the precision required for use in this population (mean difference 4.28 ± 10.94% p = 0.039 SED = 2.11; mean difference 4.44 ± 9.03% p = 0.02 SED = 1.74 respectively). The high variability in the differences contributes to an increased SED, which in turn may explain why the Wilcoxon test did not detect statistically significant deviations from zero (Table 4).

Additionally, the Bland Altman analysis showed clinically significant biases between 4.28 to 6.25%, wide limits of agreement (between 17% to 29,5%) for all of the equations and very poor absolute agreement with DEXA measurements (ICCa between −0.27 to 0.35). Also, the RMA regression showed that these three equations have substantial limitations. The Lee RC equation notably underestimated LP (−6.25%) and exhibited a negative correlation with DEXA (r = −0.13) and an inverse slope (−1.51), indicating a flawed relation. The Poortsman and ISAK 5 components equations also underestimated LM, with low correlations and excessive steep slope. Given these limitations, none of the evaluated equations are recommended for estimating LP in athletes with unilateral lower-limb amputation.

## Concurrent validity and reliability of lean mass equations

Eleven of the thirteen equations demonstrated a strong correlation with DEXA, supporting their validity for estimating LM in athletes with unilateral lower-limb amputation: De Rose and Guimaraes (r = 0.87), Lee RC 1 (r = 0.79), Lee RC 2 (r = 0.78), Douple (r = 0.78), ISAK 5 components model (r = 0.85), Olshvang (r = 0.76), Chien (r = 0.74), Lee DH (r = 0.84), Kulkarni equation 3 (r = 0.82), Kulkarni equation 4 (r = 0.83) and Salamat (r = 0.71). Additionally, all thirteen equations exhibited an almost perfect level of relative agreement with values obtained through DEXA. The five equations with the best relative agreement, considering the 95% confidence intervals are: Lee DH (ICCc = 0.95; 95% CI: 0.89 to 0.98), Kulkarni equation 4 (ICCc = 0.95; 95% CI: 0.88 to 0.96), De Rose and Guimaraes (ICCc = 0.93; 95% CI: 0.85 to 0.97), ISAK 5 components model (ICCc = 0.93; 95% CI: 0.85 to 0.97) and Salamat (ICCc = 0.91; 95% CI: 0.81 to 0.96). On the other hand, Olshvang (ICCa = 0.91; 95% CI: 0.8 to 0.96), Chien (ICCc = 0.87; 95% CI: 0.72 to 0.94), Lee DH (ICCa = 0.94; 95% CI: 0.87 to 0.97), Kulkarni equation 3 (ICCa = 0.84; 95% CI: 0.65 to 0.93) and Salamat (ICCa = 0.89; 95% CI: 0.69 to 0.96) equations showed almost perfect absolute agreement compared to DEXA. From those, only the equations from Olshvang (mean difference −1.16 ± 5.49 Kg p = 0.058 SED = 1.06), Chien (mean difference 1.37 ± 6.04 Kg p = 0.755

SED = 1.16), Lee DH (mean difference −1.52 ± 4.11 Kg p = 0.008 SED = 0.79) and Kulkarni equation 3 (mean difference 0.44 ± 5.9 Kg p = 0.683 SED = 1.14) showed non-statistically significantly difference with DEXA.

Similarly to what we found in FP estimation, these four equations again exhibited high standard deviations of the differences compared to DEXA, indicating substantial variability in their estimates. This variability resulted in large standard errors of the mean difference, which may explain why the Wilcoxon signed-rank test did not detect a statistically significant difference from zero. However, the equation Lee DH presented a lower standard deviation of the differences compared to the equations by Olshvang, Chien, and Kulkarni, resulting in a small SED (0.79).

Regarding the Bland-Altman analysis, four equations produced results broadly comparable to those of DEXA: Olshvang (Fig 10), Chien (Fig 11), Lee DH (Fig 12) and Kulkarni equation 3 (Fig 13). The Olshvang and Lee DH equations yield similar results, both slightly overestimate LM in compared to DEXA with mean differences of −1.16 Kg and −1.52 Kg, respectively. Their 95% LoA were also similar, ranging approximately from −9 Kg 7 Kg, indicating moderate variability. The Chien equation overestimated LM by 1.37 Kg, but exhibited wider LoA (−10.47 Kg to 13 Kg), suggesting less consistency in individual estimates. Notably, Kulkarni equation 3 showed the smallest difference in the estimation of LM compared to DEXA (0.44 kg), indicating minimal systematic bias. However, it wide LoA (−11.12 to 12 Kg) reflects substantial individual variability. Therefore, despite showing acceptable group-level agreement, none of these equations demonstrate sufficient precision at the individual level to be recommended for clinical use in athletes with unilateral lower-limb amputation.

In the comparison of LM estimates with DEXA using the RMA regression, the equations by Lee DH, Olshvang, and Chien, in addition to presenting minimal differences, also showed acceptable slopes (1.07 to 1.15), and moderate to high correlations (r = 0.74 to 0.84). The ISAK 5 component model and Kulkarni equation 3 also demonstrated high correlations (r > 0.8), but their steeper slopes (1.18 and 1.65) and systematic bias suggest the need for re-calibration before application. In particular, although Kulkarni equation 3 showed the least bias in mean LM, its steep slope (1.65) and large negative intercept (−31.4) indicate that it systematically over or underestimates LM across the range values, especially at extreme values. In contrast, the Heymsfield, Lee RC 2, and Janmahasatian equations showed large discrepancies, and

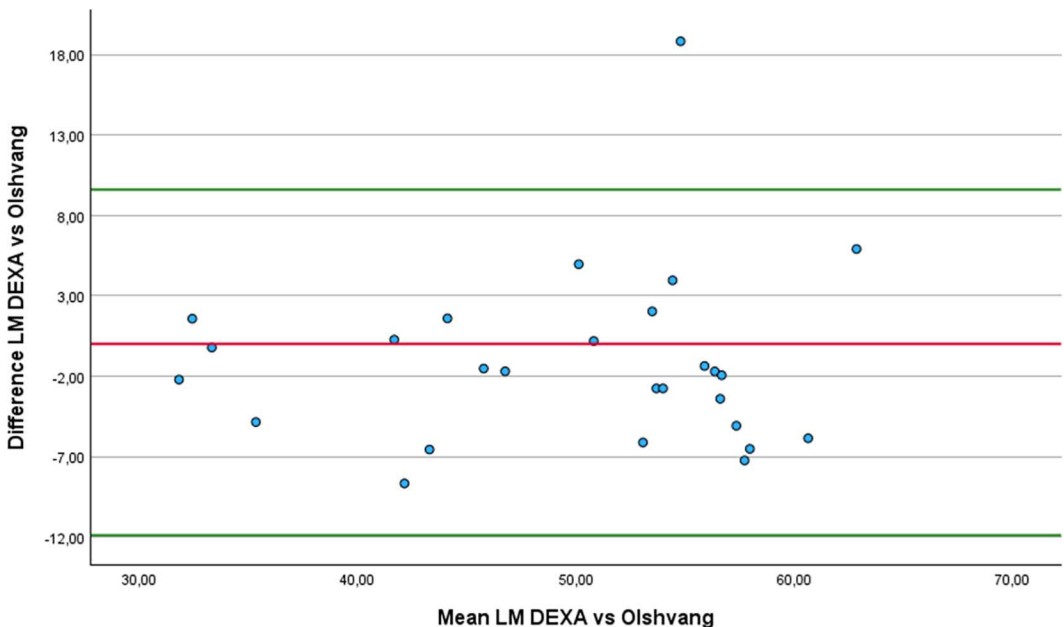

**Fig 10. Bland-Altman plot comparing LM estimates between DEXA and Olshvang equation.**

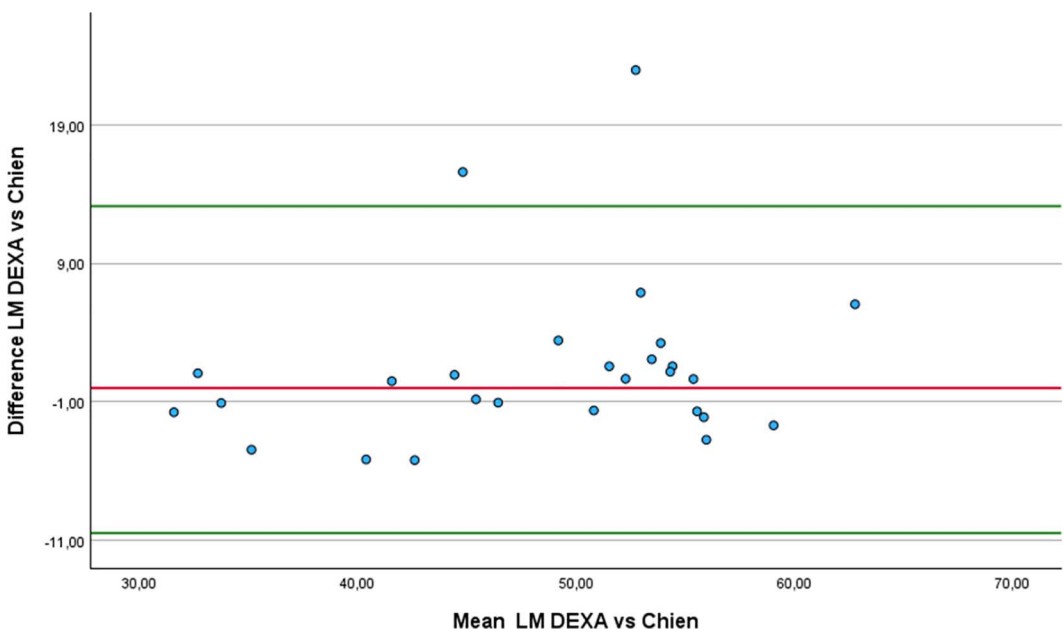

**Fig 11. Bland-Altman plot comparing LM estimates between DEXA and Chien equation.**

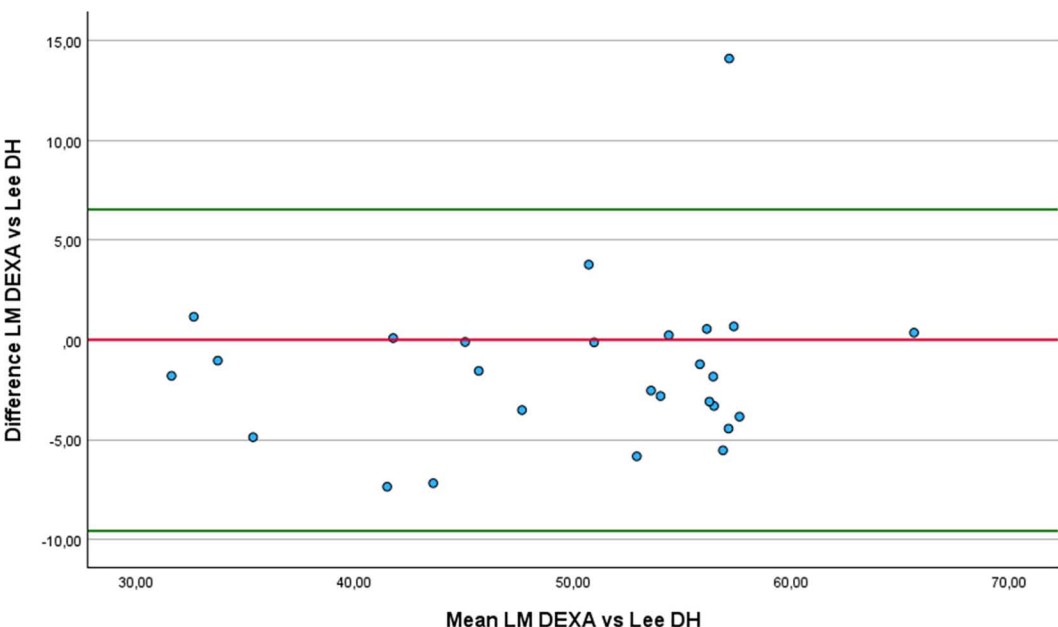

**Fig 12. Bland-Altman plot comparing LM estimates between DEXA and Lee DH equation.**

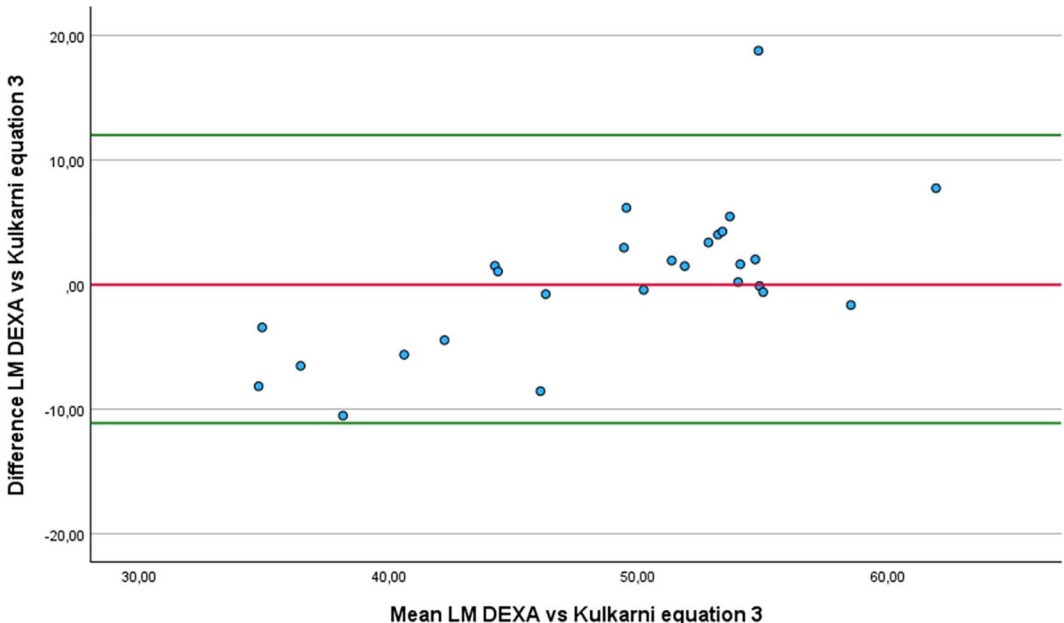

**Fig 13. Bland-Altman plot comparing LM estimates between DEXA and Kulkarni equation 3..**

inappropriate slopes, limiting their validity. Therefore, we recommend using the Lee DH [50], Olshvang [62], Chien [63] to estimate the LM in athletes with unilateral lower-limb amputation. Table 5 summarizes the most relevant equations recommended for estimating FP, FM and LP in athletes with unilateral lower limb amputation, according to the validity and reliability analysis. Also, Fig 1 Bland-Altman plots comparing the anthropometric equations with DEXA, shows the Bland Altman plots from the equations whose results were not statistically different from DEXA.

## Discussion

The present study aimed to analyze the concurrent validity of anthropometric equations for estimating body composition in athletes with unilateral lower-limb amputation compared with DEXA. For FP, our results shows that 11 out of 32 equations demonstrated a very strong correlation with DEXA, supporting their validity to measure FP. However, beyond correlation, it is essential to consider not only the validity but also the reliability and predictive accuracy of these equations in this specific population. For FP, all equations underestimated this variable compared to DEXA. This systematic underestimation is important to highlight, as DEXA measures total body fat, including region such as the head, which are often not considered by the anthropometric equations. This partly explain the observed differences and suggest limitations of these equations to fully capture whole-body fat distribution. Ten out of 32 equations showed no statistically significant differences compared to DEXA, and only three of those six equations demonstrated almost perfect absolute agreement with this instrument. That is, while many equations are available to estimate FP in athletes, only a limited number are valid and reliable for athletes with unilateral lower-limb amputation. Therefore, it is important to select the most valid and accurate method, thereby contributing equitably to the athletic process in this population.

Our results show that a classical and widely used equation, such as Durnin and Womersley with Siri, is valid for estimating FP in adults [73], which is consistent with the almost perfect relative agreement found when compared with the gold-standard method – DEXA. The present study finds an ICCa of 0.91 (95% CI: 0.81–0.96), similar to the results reported by Cavedon V. et al. [20], who determinate ICC: 0.83 agreement. Cavedon V. et al. also found that this equation

**Table 5. Most relevant equations recommended for estimating FP, FM and LP in athletes with unilateral lower limb amputation.**

| Equation to estimate fat percentage | Bias compared to DEXA (%) | Standard Error of the Difference SED (%) | ICC (IC 95%) Absolute agreement | ICC (IC 95%) Consistency | Pearson or Spearman correlation coefficient | Sample in which the equation was validated | Variables included in the equation |
|---|---|---|---|---|---|---|---|
| Hastuti [42] | 0.7 | 0.88 | 0.86 (0.68; 0.93) | 0.85 (0.68; 0.93) | 0.75 | Indonesian adults (male and females) of Javanese ethnicity living in Yogyakarta Special District Province | Triceps and suprailiac skinfold, and sex. |
| ISAK 5 components model [48,49] | 0.32 | 0.92 | 0.83 (0.62; 0.92) | 0.82 (0.61; 0.92) | 0.72 | Males and females age 6–77 years | Triceps, subscapular, suprailiac, abdominal, thigh and calf skinfolds, height and body weight. |
| Durnnin and Womersley [26] + Siri | 2.6 | 3.77 | 0.88 (0.61; 0.95) | 0.91 (0.81; 0.96) | 0.82 | Men and women aged from 16 to 72 years | Triceps, biceos subscapular and suprailiac skinfolds. |
| **Equation to estimate fat mass** | **Bias compared to DEXA (Kg)** | **Standard Error of the Difference SED (Kg)** | **ICC (IC 95%) Absolute agreement** | **ICC (IC 95%) Consistency** | **Pearson or Spearman correlation coefficient** | **Sample in which the equation was validated** | **Variables included in the equation** |
| ISAK 5 components model (47,48) | −0.71 | 0.7 | 0.89 (0.76; 0.95) | 0.89 (0.76; 0.95) | 0.81 | Males and females age 6–77 years | Triceps, subscapular, suprailiac, abdominal, thigh and calf skinfolds and height. |
| Lee DH (49) | 1.04 | 0.84 | 0.85 (0.66; 0.93) | 0.85 (0.67; 0.93) | 0.7 | Men and women with a mean age of 42.7 and 45.3 respectively. | Age, heigh, body weight, waist circumference, arm circumference, calf circumference, thigh circumference, triceps and subscapular skinfolds, race. |
| **Equation to estimate lean percentage** | **Bias compared to DEXA (%)** | **Standard Error of the Difference SED (%)** | **ICC (IC 95%) Absolute agreement** | **ICC (IC 95%) Consistency** | **Pearson or Spearman correlation coefficient** | **Sample in which the equation was validated** | **Variables included in the equation** |
| Lee DH [50] | −1.52 Kg | 0.79 | 0.94 (0.87; 0.97) | 0.95 (0.89; 0.98) | 0.84** | Men and women with a mean age of 42.7 and 45.3 respectively. | Age, heigh, body weight, waist circumference, arm circumference, calf circumference, thigh circumference, triceps and subscapular skinfolds, race. |
| Olshvang [62] | −1.16 Kg | 1.06 | 0.91 (0.8; 0.96) | 0.91 (0.79; 0.96) | 0.76** | Men and women between 45–75 years. | Age, height, body weight, waist circumference and race |
| Chien [63] | 1.37 Kg | 1.16 | 0.87 (0.72; 0.94) | 0.87 (0.72; 0.94) | 0.74** | Adults (men and women) older than 50 years old. | Body weight, sex, BMI, forearm circumference, hip circumference and sex. |

slightly overestimate FP compared to DEXA by 0.51%; in our study, we found a larger overestimation (2.6%). But the slope of 0.74 and an intercept of 6.22 indicates proportional bias; it underestimates higher FP and overestimates lower ones. They also found narrower LoA compared to what we found in our study (−3.84 to 4.86) [20].

On the other hand, we found that the Forsyth and Sinning equations did not show statistically significant differences from DEXA (Siri, p = 0.642; Brozek 0.342), contrary to the findings of Cavedon et al. [20], who found a significant bias of −1.51% (p < 0.001). They also found a strong correlation (r = 0.88), a slope close to 1 (0.94), a near zero intercept (−0.23) and acceptable Loa (−4.45 to 1.42). However, the absolute and relative agreements of these two equations with DEXA reported in our study were substantial, (r = 0.55 for both). Also, the low slopes (0.63 and 0.68) and high intercepts (10.6 and 9.8) indicate substantial proportional and fixed bias, meaning that those equations tend to underestimate FP in lean individuals and overestimate in those with higher FP. Additionally, the high standard deviations suggest a wide variability in estimates. Therefore, despite producing mean values close to those obtained by DEXA, the Forsyth and Sinning method demonstrate poor agreement and should not be recommended for precise individual or clinical assessment in athletes with unilateral lower-limb amputation.

Regarding the White (with Siri and Brozek), Katch and McArdle (with Siri and Brozek), Nagamine and Suzuki (with Siri and Brozek), Wilmore and Behnke (with Siri and Brozek), Evans and Eston equations, we found results similar to those of Cavedon et al. [20]. The bias compared to DEXA was statistically significant (between −2.77 to −8.03%) and the LoA were wide (range from 8.13 to 10.39). Therefore, we do not either recommend any of these equations for the estimation of FP in athletes with unilateral lower-limb amputation, even though the correlations with DEXA could be moderate to high.

Our study found that the equation proposed by Hastuti [42] was a valid option for estimating FP in athletes with lower-limb amputation, even though this equation was validated for the Indonesian population. Regarding the reliability, the regression analysis comparing the Hastuti equation to DEXA revealed a slope o. 1.16, and an intercept of −3.6, indicating proportional bias in which this equation tends to overestimate FP in individuals with higher adiposity and underestimate in leaner individuals. The Bland-Altman analysis further supports this interpretation, showing wide LoA (−8.2 to 9.6%) and the presence of two outliers; these outliers reduce confidence in the method´s reliability by highlighting substantial individual discrepancies, potentially due to measurement error or variation in body composition not accounted for by the equation. To the best of our knowledge, no cross-validation of this equation has been published for athletes with unilateral lower-limb amputation that would allow us to compare our results.

Another equation that showed excellent validity in estimating FP in athletes with unilateral lower-limb amputation compared with DEXA was the Lee HD [50]. This equation was validated in a large sample of individuals from the United States, including 2015 women and 2292 men. The authors proposed four different equations for estimating FP in men and women; however, in this study, we applied the equation with the highest coefficient of determination ($R^2$ 0.80 for men and 0.74 for women). These equations incorporated eight anthropometric variables (weight, height, circumference, and skinfold) and accounted for sex, age, and race. The authors also reported that these equations were derived from a heterogeneous population, making them robust in estimating FP in populations with diverse characteristics [50]. On the other hand, the RMA regression showed a slope close to 1 (0.96), and a small intercept (2), indicating limited proportional bias and reasonably fit overall. However, the wide LoA (−9.8 to 11.7) suggest substantial individual variability, which may impact its reliability.

Comparing Deborah-Kerr anthropometry equations [48] (used by ISAK to determinate five components) with DEXA, we found a minor non-statistically significant overestimation of FP by 0.32 Kg (95% CI −1.58 to 2.22) and an almost perfect relative ICCc = 0.82 and absolute ICCa = 0.83 agreement, which make this equation a very good option to apply in athletes with unilateral lower-limb amputation. We also found a slope of 1.26 that indicates a tendency to overestimate FP in individuals with higher adiposity and underestimate it in those with lower values, showing a pattern of proportional bias. This bias may be partly explained by the reduced reliability of skinfolds measurements in individuals with obesity, due to difficulties in grasping and compression thick subcutaneous fat, limited caliper capacity, and challenges in identifying anatomical landmarks [74]. Also, the intercept of −6.6 reflects a systematic underestimation of FP compared to DEXA.

The procedures, methods, and equations have been widely used and fully standardized by the International Society for the Advancement of Kinanthropometry (ISAK), which is dedicated to the rigorous training of professionals responsible for conducting these measurements [75]. These equations have been used to estimate the body composition of elite male and female soccer players [76], young professional soccer players [77], and elite judo athletes [78]. It is recommended that measurements be conducted by an ISAK-certified professional at level 2 or 3, enduring measurement error rates of ≤7.5% for skin folds and ≤1% for other variables. Additionally, certification at these levels qualifies professionals to conduct a complete body composition assessment, including the use of Deborah-Kerr anthropometric equations, such as the ISAK 5 component model [75]. In contrast, ISAK Level 1 certification allows only the application of the Carter, Faulkner, and Slaughter equations. Although our results showed moderate to high correlations with DEXA (r = 0.88, r = 0.82, r = 0.78, respectively), their reliability was limited. We observed substantial biases ranging from 8.81% to 13.33%, and Bland-Altman analysis revealed wide LoA, with upper bounds reaching up to 21%. These findings suggest that, despite acceptable validity, the precision of the equations is insufficient for accurate individual assessment.

Finally, we identified in the literature that Cavedon et al [20] proposed two equations for estimating FP in athletes with unilateral lower limb amputation. These were validated in a sample of 29 subjects with this condition and showed good predictive performance. Table 6 summarizes the equations along with the variables required for the estimation. Unfortunately, we did not collect axillary and chest skinfold measurements, which are necessary to calculate FP using these equations. Nevertheless, based on the results reported by Cavedon et al, we recommend the use of this method for estimating FP in this specific population.

In our study, we also found that the ISAK 5 components model (Deborah-Kerr anthropometry equations [48]) has an almost perfect relative agreement (ICCc = 0.89) and absolute agreement (ICCa = 0.89) for FM estimation, compared to DEXA, and FM results were not statistically different from this gold standard (p = 0.323). In terms of agreement, our results are in accordance with those reported by Arias Téllez MJ et al. [79], who determined a 95% LoA of 9.95 and 4.04 kg to calculate LM for this equation. On the other hand, no study has validated the Al-Guindan equations [52] on athletes using DEXA as a reference. Notably, our findings highlight an almost perfect absolute agreement (ICCa: 0.83, 95% CI: 0.62–0.92), with no statistically significance difference (p = 0.431) of the results between the methods for determinate FM on athletes with unilateral lower-limb amputation. In contrast, the equations from Lee DH [50] demonstrated validity and reliability for estimating FM in this population. They reported that the equation derived from men and women achieved a determination coefficient of 0.93 and 0.94, respectively, which makes it a good option to estimate LM in athletes with lower-limb amputation. Furthermore, regarding the reliability of these equations, the ISAK 5 components model showed the best overall agreement with DEXA. Besides the strong correlation, it showed minimal systematic bias (0.71 Kg), and relatively narrow LoA (−7.8 to 6.4), indicating both accuracy and consistency in individual estimates. The Lee DH and Al-Guidan equations also presented small mean differences from DEXA and moderate correlations, suggesting acceptable performance. In contrast, although De Rose and Guimaraes equation had the highest correlation (r = 0.85),

**Table 6. Equations proposed by Cavedon et al to estimate fat percentage in athletes with unilateral lower lib amputation.**

| Equations proposed by Cavedon et al [20] | | Variables required for the application of the equation | Mean determination coefficient R² and range | Mean absolute prediction error and range |
|---|---|---|---|---|
| Cavedon 1 | FP = 0.241 * (TH) + 0.418 * (AB) + 0.329 (SS) + 0.259 (AX) − 1.689 | Anterior thigh, abdominal, subscapular and axillary skinfolds | 0.89 (0.78–0.94) | 1.18 (0.85–1.73) |
| Cavedon 2 | FP = 0.162 * (BI + TR + SS + AX + CH + SI + AB + TH + C) − 0.311 | Bicipital, tricipital, subscapular, axillary, chest suprailiac abdominal anterior thigh and calf skinfolds. | 0.92 (0.9–0.93) | 1.03 (0.96–1.18) |

Skinfolds (SK): BI, bicipital; TR, tricipital; SS, subscapular; SI, suprailiac; AB, abdominal; TH, thigh; C, calf; AX axillar; CH chest.

it substantially underestimated FM (6.45 Kg) and had a wide LoA (−0.2 to 13.1), limiting its practical applicability. The Salamat and Heitmann equations exhibit both lower correlations and wider LoA, indicating poorer agreement with DEXA to estimate FM in athletes with unilateral lower limb-amputation.

Regarding the LP, all equations showed statistically significant differences from DEXA (p < 0.05) and very low absolute agreement compared with the gold standard. The accuracy of these equations for predicting LM and LP has been evaluated in the general population, and similar results have been obtained. Baglietto et al. [80] concluded that there were statistically significant differences between the equations proposed by Kerr [48] (p < 0.001), Lee [57] (p < 0.001), Heymsfield (p < 0.001) [59] and DEXA. However, they also reported that the only equation with results that were not significantly different from those of DEXA was that proposed by Poortsman [58]. This is interesting since, in our study, this equation exhibits the least bias compared to DEXA (mean difference 4.28 Kg ± 10.94 p = 0.039). Also, none of the equations demonstrated strong agreement with DEXA. The ISAK 5 components model demonstrated proportional bias (slope = 1.93) and a large intercept (−38.3), indicating systematic overestimation at lower values and underestimation at higher values. The Poortsman equation had similarly low correlation and a slope of 1.48, with a negative intercept (−17.9), suggesting poor agreement and notable bias. The Lee RC equation showed a negative correlation (r = −0.13) and highly divergent slope (−1.51) with an extreme intercept (117.8), indicating and inverse and implausible relation with DEXA. Therefore, these results suggest that none of the equations provide valid estimates of LP in this sample, highlighting the limitations of using anthropometric methods for this specific body composition parameter in athletes with lower-limb amputation.

For the estimation of LM, three equations were demonstrated to be valid and reliable for application in athletes with unilateral lower-limb amputation: Lee DH [50], Olshvang [62], and Chien [63]. The Lee DH equation was developed and validated in a large sample of 5239 men and 4519 women, demonstrating strong predictive performance for estimating both LM and FM. This practical model incorporated age, race, heigh, weight, and waist circumference, and achieved high coefficients of determination for LM (men: $R^2 = 0.91$, standard error of the estimate (SEE) = 2.6 kg; women: $R^2 = 0.85$, SEE = 2.4 kg) and fat mass (men: $R^2 = 0.90$, SEE = 2.6 kg; women: $R^2 = 0.93$, SEE = 2.4 kg) [50]. The equation from Olshvang employed machine learning algorithms to estimate LM based on demographic variables and DEXA data. The primary dataset was the National Health and Nutrition Survey (NHANES), covering the years 1999–2018 and including a representative sample of individuals aged 45–75 years. However, the authors did not report the exact sample size. Despite this, the approach demonstrated promise, though the authors acknowledge the need for further validation in more diverse populations [62]. The Chien equation was developed using a sample of 332 participants and validated in an additional 166 individuals. It demonstrated strong predictive performance, with a coefficient of determination of $R^2 = 0.9$ and a standard error of the estimate of 0.31 Kg. Although Kulkarni equation 3 showed the smallest mean difference compared to DEXA (0.44 Kg ± 5.9), and the absolute and relative agreement were almost perfect, its pronounced slope and substantial negative intercept (−31.4) suggest that this equation overestimates or underestimates LM, with greater discrepancies occurring at extremes. The equations from Chien and Olshvang demonstrated good reliability and validity for application in our population, becoming a great option over more classic equations to estimate LM. Their ability to estimate LM using accessible anthropometric variables, such as sex, weight, height, skin folds, and circumferences, makes them especially suitable for applied setting.

Finally, it is important to clarify the difference between Appendicular Muscle Mass (AMM), Lean Mass (LM) and Fat-Free Mass (FFM); even though they are related, are distinct body composition components. FFM includes all nonfat tissues in the body, such as muscles, bones, water, and organs [13,81]. LM is often interchangeable with FFM [43,55,69] but may sometimes include a small amount of essential fat [81]. AMM specifically refers to limb muscle mass and is a key indicator of skeletal muscle health. Understanding these differences is crucial when assessing body composition because different equations and methods may estimate them differently. Therefore, several equations have been developed to estimate appendicular muscle mass (AMM) and lean mass (LM). Specifically, the equations proposed by De Rose and Guimaraes [53], Heymsfield [53,59], Lee 1 [57], Lee 2 [57], Douple [60], and the ISAK 5-component model [48,49] were

designed to estimate the AMM. In contrast, the equations of Janmahasatian [61], Olshvang [62], Chien [63], Lee DH [50], Kulkarni equation 3 [64], and Salamat [56] were used to estimate LM.

This study employed a reference method (DEXA) for the estimation and comparison of body composition, and efforts were made to minimize measurement in the anthropometric estimates by involving certified anthropometrist, which constitutes a methodological strength. However, several limitations should be acknowledged. Due to the specificity of the study population (Paralympic athletes with unilateral lower-limb amputation) the sample size was relatively small and comparisons between sports modalities were not possible because of the low statistical power. On the other hand, measurements were conducted following the ISAK protocol and some specific anthropometric variables required by certain equations, such as axillary skinfold and wrist breadth, were not collected. As a result, recently developed equations for general heterogeneous population such as the one proposed by Rojano-Ortega et al. [82] and tailored for individuals with unilateral lower-limb amputations, such as those proposed by Cavedon et al. [20], could not be included in the analysis. Instead, the equations applied in our study were originally validated in able-bodied populations. To our knowledge, while Cavedon et al. have evaluated the validity of some of these equations in individuals with limb loss, none have been specifically validated for athletes with amputation.

In this study, anthropometry, recognized as a doubly indirect method for estimating body composition, demonstrated significant agreement with more direct techniques such as DEXA and BIA, despite the inherent differences among these methods [16,83]. Its widespread application in sports settings is attributed to its accessibility, cost-effectiveness, and utility for monitoring training adaptations, predicting performance, and assessing the physical and nutritional status of the athlete. These advantages highlight its values, not only in general athletic population but also in more specialized groups, such as athletes with limb amputations [20,83]. Anthropometric methods, which involve measurements such as skinfolds, height, and girths, are used in predictive equations to estimate tissue components. However, these equations are typically developed and validated in specific populations, making the non-interchangeable across different demographic or athletic groups [19,84]. This highlights the need to develop anthropometry-based predictive equations specifically tailored to athletes with lower-limb amputation, following standardized measurements protocols such as those established by the ISAK. Adherence to those guidelines can help ensure best practices in applied sports settings [19,84]. One notable limitation in the current body of research is the small sized used for validation. For instance, Cavedon et al. [20] validated their equation in only 24 subjects, while the present study included 27 participants. To address this, future efforts should focus on fostering international collaboration among researchers working with athletes with unilateral lower-limb amputation. Such collaboration would allow for the pooling of data, increasing statistical power and enabling the development and validation for more robust and generalizable predictive equations. Variability in predictive performance can arise from differences in the reference methods used during equations development and the characteristics of the validation sample, including sex, sport type, and geographical origin [84]. Therefore, selecting an appropriate equation requires careful consideration of these contextual factors.

## Conclusion

Despite the absence of previous studies that have validated anthropometric equations to estimate body composition in athletes with unilateral lower-limb amputation, the results of the present study provide relevant information on a low-cost and less technically complex method for measuring FP, FM, LP, and LM in athletes with unilateral lower-limb amputation. The selection of equations and their interpretation should consider the degree of error estimation inherent in this method and some adjustments owing to the absence of body parts. For this population, the following recommendations are made:

For athletes with unilateral lower-limb amputation, we recommend Hastuti [72], the ISAK 5 components model [48,49] and Durnnin and Womersley with Siri [26] to estimate FP. For the estimation of FM, we recommend the ISAK 5 components model [48,49], followed by Lee DH [50] equations. For LM, we recommend the Lee DH [50], Olshvang [62], or Chien [63]. Finally, we do not recommend the use of any of the Lee [57], Poortsman [58], or ISAK 5 components model [48,49]

equations to estimate LP. Athletes with unilateral lower-limb amputation deserve to have their own validated methods to estimate their body composition, not only to contribute to their performance but also their health. Therefore, we encourage researchers and professionals to validate the specific evaluation methods for this population.

## Supporting information

**S1 Data.**
(XLSX)

## Acknowledgments

We thank Dr Miguel Alexander Niño M.D and Nutritionist Guillermo Antonio López López for their support in managing the logistics necessary to proceed with this project. Additionally, we acknowledge the support of the Applied Sciences Unit (UCAD) from the District Institute of Recreation and Sport (IDRD) form Bogotá, Colombia for providing access to the population. The authors declare no competing interests.

## Author contributions

**Conceptualization:** Laura Victoria Rivera-Amezquita, Ximena Saavedra-Bernal, Diana Carolina Escorcia-Gomez, Diana Marcela Ramos-Caballero.

**Data curation:** Laura Victoria Rivera-Amezquita, Ximena Saavedra-Bernal, Sofia Diaz-Moreno.

**Formal analysis:** Laura Victoria Rivera-Amezquita, Ximena Saavedra-Bernal.

**Funding acquisition:** Laura Victoria Rivera-Amezquita.

**Investigation:** Laura Victoria Rivera-Amezquita, Ximena Saavedra-Bernal, Sofia Diaz-Moreno, Alejandra Tordecilla-Sanders, Diana Carolina Escorcia-Gomez, Diana Marcela Ramos-Caballero.

**Methodology:** Laura Victoria Rivera-Amezquita, Ximena Saavedra-Bernal, Sofia Diaz-Moreno, Alejandra Tordecilla-Sanders, Diana Marcela Ramos-Caballero, Zdenek Svoboda.

**Project administration:** Laura Victoria Rivera-Amezquita, Alejandra Tordecilla-Sanders, Diana Marcela Ramos-Caballero.

**Resources:** Laura Victoria Rivera-Amezquita, Diana Marcela Ramos-Caballero, Zdenek Svoboda.

**Software:** Laura Victoria Rivera-Amezquita.

**Supervision:** Diana Carolina Escorcia-Gomez, Zdenek Svoboda.

**Validation:** Laura Victoria Rivera-Amezquita, Sofia Diaz-Moreno, Zdenek Svoboda.

**Writing – original draft:** Laura Victoria Rivera-Amezquita, Ximena Saavedra-Bernal, Alejandra Tordecilla-Sanders.

**Writing – review & editing:** Laura Victoria Rivera-Amezquita, Sofia Diaz-Moreno, Diana Carolina Escorcia-Gomez, Diana Marcela Ramos-Caballero, Zdenek Svoboda.

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
