## [Decision Letter · Decision Letter 0]

PONE-D-25-11488Validity and reliability of anthropometric equations versus Dual X-ray Absorptiometry to estimate body composition in athletes with unilateral lower-limb amputationPLOS ONE

Dear Dr. Rivera-Amezquita, 

Thank you for submitting your manuscript to PLOS ONE. After careful consideration, we feel that it has merit but does not fully meet PLOS ONE’s publication criteria as it currently stands. Therefore, we invite you to submit a revised version of the manuscript that addresses the points raised during the review process. 

We look forward to receiving your revised manuscript.

Kind regards,

Alberto Souza Sá Filho, Ph.D

Academic Editor

PLOS ONE

Journal Requirements:

“This research was funded by Universidad del Rosario (Specific grant number: IV-FCS024 Capital Semilla)”

3. Please note that funding information should not appear in the Acknowledgments section or other areas of your manuscript. We will only publish funding information present in the Funding Statement section of the online submission form. Please remove any funding-related text from the manuscript. 

**Additional Editor Comments:**

Dear authors, this manuscript presents significant value in the area in question, and was considered to continue the review process. Nevertheless, numerous adjustments are still required to ensure the delivery of the best final product. Kindly take into consideration the comments provided by the three reviewers who devoted their time to support the improvement of your manuscript. We believe that 2 weeks would be sufficient time to complete your manuscript. However, if you need more time, please let us know. Thank you for submitting to PLOS ONE.

Reviewers' comments:

Reviewer's Responses to Questions

**Comments to the Author**

1. Is the manuscript technically sound, and do the data support the conclusions?

Reviewer #1: Partly

Reviewer #2: Yes

Reviewer #3: Partly

2. Has the statistical analysis been performed appropriately and rigorously? 

Reviewer #1: Yes

Reviewer #2: Yes

Reviewer #3: No

3. Have the authors made all data underlying the findings in their manuscript fully available?

Reviewer #1: Yes

Reviewer #2: Yes

Reviewer #3: Yes

4. Is the manuscript presented in an intelligible fashion and written in standard English?

Reviewer #1: Yes

Reviewer #2: Yes

Reviewer #3: Yes

5. Review Comments to the Author

Reviewer #1: This article is important and relevant to sports science and especially to the category of amputee athletes. It aims to create an evaluation method that is specific to this population. However, it needs revisions since it presents a lack of clarification on points that are relevant and important for the construction of knowledge.

Reviewer #2: General comments: I congratulate the authors on the paper, I believe that with only small changes, it reaches a level of publication that encompasses the scope of the newspaper.

First of all, I would like to thank the authors and editorial board for the opportunity to review their manuscript. I hope that my comments are pertinent so that we can move towards a final product of high quality that corroborates the scope of the research in question.

Title : The title is cohesive and clear with the aim of the studies, without considerations.

Abstract:

The summary is well structured, and well described, however I believe that some changes may be well liked, I hope that we can reach a consensus on my recommendations.

First, regarding the methods in line 31 "This cross-sectional study included 27 athletes (22 men and five women)", consider reporting on which type of sport your study focused, consider adding mean and standard deviation values for age. It is clear that these are athletes with some level of amputation, due to the large number of sports that this population participates in. Consider highlighting this in the abstract.

Still in the methods on line 33 "fat percentage (FP), lean percentage, lean mass (LM)," if the term ´´lean percentage`` is used during the work, I recommend including the acronym, as well as for the other terms.

I recommend including that correlation tests were carried out between the DEXA results and the equations.

Introduction

Firstly, I would like to congratulate the authors, for the rationale constructed and for the introductory explanation, I will just make 2 small suggestions that I hope you will consider. The introduction presents an excellent trigger of ideas, just as the gap in the study is evident, it is the referential framework that supports this. My first recommendation is to combine the first and second paragraphs. I believe that being more direct in the introduction seems more inviting to readers. I reiterate here my statement about writing, starting at line 70, where the limitations of using DEXA are made clear, the limitations regarding the equations, as well as the research gap. I congratulate the authors again. My second suggestion is that at the end of the introduction, the hypotheses that were raised should be clearly stated. I believe that during the construction of the investigation, as is common in research, hypotheses about the possible results were raised.

Methods

On line 103 the term ´´ lean percentage (LP) `` appears with the abbreviation, insert in the summary.

The methods are well described, as are the procedures, during reading I came up with some doubts that may be relevant and I would like to be clarified about them, regarding the anthropometric measurements it is clear that it was carried out by two researchers, with Isak certifications ´´ Anthropometric measurements were conducted by two level 1 anthropometrists following the restrictive profile protocols of the International Society for the Advancement of Kinanthropometry (ISAK)´´, two weeks of training periods are reported, prior to this, a cohesive conduct to be carried out, the authors declare that while one evaluator carried out the measurements, the other took notes, so no type of blinding was carried out so that the evaluators were not aware of the measurements carried out, if any type of blinding was carried out, I think it is worth reporting.

Sample size

A justification for the sample size is valid, if an a priori sample calculation was made, or a more detailed justification for the sample size.

Statistical analysis

The statistical procedures are well described. I congratulate the authors for presenting the data with confidence intervals. I would like to draw attention to a small detail: in the analyses, the correlations were established using the Pearson and Spearman tests. However, in the statistical analyses, only the Spearman Correlation is described. It is only when we look at the results in Table 4 that it is presented. No further comments on the analyses.

Results

No comments on the results, they are well described, followed by tables and significance values.

Discussion

Without further considerations for the discussion, I congratulate the authors for presenting the limitations, as well as recommendations for the use of the equations, if they adopt in the introduction to present the hypotheses raised, it is recommended to reiterate whether these were accepted, or refuted.

Reviewer #3: Abstract:

The author should add "(" (line 43)

The authors reach a questionable conclusion in lines 48-50, stating "Based on our findings, for athletes with unilateral lower-limb amputation, we recommend Hastuti, the ISAK 5 components model, and Lee to estimate FP." However, in lines 52 and 53, the authors state, "Finally, we do not recommend the use of any of the Lee, Poortsman, or 5 component ISAK equations to estimate FP." To ensure clarity and coherence for the reader, the authors should present a direct conclusion that succinctly synthesizes the salient findings of the study.

Introduction:

The references are repeated. As a suggestion, the following arrangement of the references could be considered: (13-15,23-25) (Line 77).

Methods:

The authors do not clearly describe the characteristics of the participants in this section. The depiction of the subjects' profiles is of high importance in this study, given that only in the results section are different sports disciplines presented, which suggests that some of them may be amputees and paraplegics while others are not. This potential variability could have substantial implications for the study's outcomes (Lines 182-187).

Dual-Energy X-ray Absorptiometry: The authors did not provide a comprehensive description of the equipment calibration process (DEXA); they merely indicated the utilization of an anthropomorphic spine phantom. It is imperative to ascertain the number of assessments conducted prior to the calibration of the equipment. Furthermore, the test-retest value of the device with the anthropometric spine phantom must be elucidated. Additionally, the reference cited is specific to men, but the study was also carried out with women. Consequently, a similar reference with women would be pertinent (line 133).

Body Composition Measurements through Anthropometry: While the authors indicated that the measurement variation between evaluators was consistent with the ISAK standard, they did not furnish data to substantiate these values. The demonstration of the ICC of the measurement between evaluators and the stability of the measurement is essential, given the purpose of the study (line 150).

Statistical analysis: The authors of the study describe the distribution as either parametric or non-parametric. However, it is imperative to note that the distribution in question can only be normal or non-normal, and the statistical analysis can be parametric or non-parametric (Line 158). Additionally, The authors did not mention Pearson's correlation coefficient. Moreover, there was no discussion of how outliers were treated, since their presence may introduce a greater degree of correlation bias between variables. These methodological limitations are evident in the Bland-Altman data, which demonstrates a lack of precision and reliability (Line 170).

Results:

The authors provide a superficial description of the characteristics of the participants. However, the appropriate section for this would be the methods section.

The authors could present other dispersion variables in those tables (tables 2 and 3) that better describe the data, such as maximum, minimum, and coefficient of variation.

Concurrent validity and reliability of lean percentage: Skip a line on the page (line 265).

Discussion and Conclusion:

The authors' discussion and conclusions are based on the presentation of comparisons obtained between the anthropometric equations and DEXA. However, it is imperative to acknowledge the potential impact of outliers on the outcomes, irrespective of the employed equation or variable. The correlation coefficient and paired t-test analyses are particularly susceptible to the presence of outliers, thereby compromising the reliability and validity of the results.

Acknowledgements: Skip a line on the page (line 454).

6. PLOS authors have the option to publish the peer review history of their article (what does this mean? ). If published, this will include your full peer review and any attached files.

**Do you want your identity to be public for this peer review?** For information about this choice, including consent withdrawal, please see our Privacy Policy .

Reviewer #1: **Yes: ** Bueno, JCA

Reviewer #2: **Yes: ** Pedro Augusto Inacio

Reviewer #3: **Yes: ** Anderson Luiz Bezerra da Silveira

---

## [Author Response · Author response to Decision Letter 1]

23 May 2025

Dear Reviewers,

We sincerely thank you for taking the time to conduct a thorough and thoughtful review of our manuscript. Your insightful comments and suggestions have been invaluable in helping us improve the quality and clarity of our work.

As a team, we carefully addressed each of your observations and did our best to fully meet the requirements outlined in your feedback. In response, we strengthened the data analysis procedures (we use a Reduced major axis (RMA) and calculate the Bland-Altam Limits of agreement for all of the equations), which allowed us to deepen the interpretation and discussion of our findings. Notably, we incorporated a more detailed analysis of the influence of outliers on the reliability and agreement of the evaluated equations. This addition provided a clearer understanding of the robustness and limitations of the methods used.

We are confident that the revised manuscript better reflects the scientific rigor expected by the journal, and we greatly appreciate your contribution to this improvement process.

Thank you once again for your constructive input and the opportunity to revise our work.

Laura Victoria Rivera Amézquita

Principal Professor

Universidad del Rosario, School of Medicine and Health Sciences, Rehabilitation Science Research Group, Center for the Study of Physical Activity Measurement (CEMA), Bogotá 111211, Colombia.

Carrera 24 # 63c-69. Universidad del Rosario, Bogotá, Colombia

Laurav.rivera@urosario.edu.co

+57 3163885488

---

## [Editor Report · Decision Letter 1]

Validity and reliability of anthropometric equations versus Dual X-ray Absorptiometry to estimate body composition in athletes with unilateral lower-limb amputation: a pilot study

PONE-D-25-11488R1

Dear Dr. Rivera-Amézquita,

We’re pleased to inform you that your manuscript has been judged scientifically suitable for publication and will be formally accepted for publication once it meets all outstanding technical requirements.

Kind regards,

Alberto Souza Sá Filho, Ph.D

Academic Editor

PLOS ONE

Additional Editor Comments:

Dear Authors,

After a thorough review of the contents, it now appears to us that the manuscript entitled "Validity and reliability of anthropometric equations versus Dual X-ray Absorptiometry to estimate body composition in athletes with unilateral lower-limb amputation: a pilot study" is suitable for publication and indeed contributes meaningfully to the advancement of scientific knowledge.

We sincerely appreciate your dedication throughout this process.

Thank you.

---

## [Editor Report · Acceptance letter]

PONE-D-25-11488R1

PLOS ONE

Dear Dr. Rivera-Amezquita,

I'm pleased to inform you that your manuscript has been deemed suitable for publication in PLOS ONE. Congratulations! Your manuscript is now being handed over to our production team.

Kind regards,

on behalf of

Dr. Alberto Souza Sá Filho

Academic Editor

PLOS ONE